# Marked T cell activation, senescence, exhaustion and skewing towards TH17 in patients with COVID-19 pneumonia

Sara De Biasi[1,9], Marianna Meschiari[2,9], Lara Gibellini[1], Caterina Bellinazzi[1], Rebecca Borella[1], Lucia Fidanza[1], Licia Gozzi[1], Anna Iannone[1], Domenico Lo Tartaro[1], Marco Mattioli[1], Annamaria Paolini[1], Marianna Menozzi[2], Jovana Milić[2], Giacomo Franceschi[2], Riccardo Fantini[3], Roberto Tonelli[3], Marco Sita[4], Mario Sarti[5], Tommaso Trenti[5], Lucio Brugioni[6], Luca Cicchetti[7], Fabio Facchinetti[1], Antonello Pietrangelo[1], Enrico Clini[3], Massimo Girardis[4], Giovanni Guaraldi[2], Cristina Mussini[2] & Andrea Cossarizza[1,8⊠]

The immune system of patients infected by SARS-CoV-2 is severely impaired. Detailed investigation of T cells and cytokine production in patients affected by COVID-19 pneumonia are urgently required. Here we show that, compared with healthy controls, COVID-19 patients' T cell compartment displays several alterations involving naïve, central memory, effector memory and terminally differentiated cells, as well as regulatory T cells and PD1 $^+$CD57$^+$ exhausted T cells. Significant alterations exist also in several lineage-specifying transcription factors and chemokine receptors. Terminally differentiated T cells from patients proliferate less than those from healthy controls, whereas their mitochondria functionality is similar in CD4$^+$ T cells from both groups. Patients display significant increases of proinflammatory or anti-inflammatory cytokines, including T helper type-1 and type-2 cytokines, chemokines and galectins; their lymphocytes produce more tumor necrosis factor (TNF), interferon-γ, interleukin (IL)-2 and IL-17, with the last observation implying that blocking IL-17 could provide a novel therapeutic strategy for COVID-19.

[1] Department of Medical and Surgical Sciences for Children and Adults, University of Modena and Reggio Emilia School of Medicine, Via Campi 287, 41125 Modena, Italy. [2] Infectious Diseases Clinics, AOU Policlinico and University of Modena and Reggio Emilia, via del Pozzo 71, 41124 Modena, Italy. [3] Respiratory Diseases Unit, AOU Policlinico and University of Modena and Reggio Emilia, via del Pozzo 71, 41124 Modena, Italy. [4] Department of Anesthesia and Intensive Care, AOU Policlinico and University of Modena and Reggio Emilia, via del Pozzo 71, 41124 Modena, Italy. [5] Clinical Microbiology Unit, AOU Policlinico, via del Pozzo 71, 41124 Modena, Italy. [6] Emergency Department, MIAC, AOU Policlinico, via del Pozzo 71, 41124 Modena, Italy. [7] Labospace, via Apelle 41, 20128 Milano, Italy. [8] National Institute for Cardiovascular Research, via Irnerio 48, 40126 Bologna, Italy. [9]These authors contributed equally: Sara De Biasi, Marianna Meschiari. ⊠email: andrea.cossarizza@unimore.it

The dramatic pandemic caused by the severe acute respiratory syndrome coronavirus (SARS-CoV-2) that causes Corona Virus Disease-2019 (COVID-19)[1], is a dramatic threat to our species that is changing daily habits and social behaviors all over the world. In Italy, the first patients with severe pneumonia of unknown origin were observed in Lombardy, first confirmed case dating to February 21st, 2020[2]. From that moment, we saw an unpredictable number of patients with severe pneumonia, whose treatment often required admission into intensive care units. Unfortunately, many patients failed in their fight against the virus, and, as of June 18th, Italy has counted more than 35,000 deaths amongst over 240,000 infections.

The pathophysiology of such virus is not yet completely understood[3,4]. Previous studies on severe diseases caused by other coronaviruses indicated that pulmonary inflammation is associated with increased plasma levels of proinflammatory cytokines[5,6]. A similar phenomenon has been described in patients with COVID-19, who can experience the so-called "cytokine storm[7]". However, little is known about the cells that are involved in the production of these mediators or about the specific immune response of patients with COVID-19.

Currently, we, and others, are observing that COVID-19 is much more severe in elderly patients, especially in those with different comorbidities, diabetes, and obesity, while it appears much less severe in children and pregnant women[8–10]. Immunological aging is associated with a chronic, subclinical inflammatory state, defined "inflammaging" where T helper (TH) type 1 lymphocytes play a crucial role, whereas children and pregnant women have a tendency to develop TH2 responses, so producing less proinflammatory molecules[11,12]. In order to better understand how immune response and cytokine production are orchestrated, we have deeply characterized T cells, and have studied the functional differentiation of T cells towards TH1, TH2, or TH17. For this purpose, we have employed an array of flow cytometric approaches combined with sophisticated techniques for unsupervised analysis of complex T cell phenotypes, and we have investigated the proliferative capability of different CD4+ and CD8+ T cell subsets, along with mitochondria functionality in CD4+ T cells. We have finally measured plasma levels of 31 cytokines and the polyfunctionality of T cells.

A retrospective study performed by our group evidenced that a drug able to block the biological activities of IL-6, such as tocilizumab, can significantly reduce invasive mechanical ventilation or death in severe COVID-19 pneumonia (defined as the concomitant presence of a respiratory rate ≥30 breaths per minute, blood oxygen saturation ≤93%, a $PaO_2/FiO_2$ ratio <300 mmHg in room air and lung infiltrates >50% of the lung, filed within 24–48 h)[13]. Even if our results need to be confirmed by randomized trials, it is to note that, in comparison with the control group, COVID-19 patients treated with tocilizumab had an increased prevalence of severe infection.

Here we show that COVID-19 patients have reduction in absolute numbers of CD4+ and CD8+ T lymphocytes, which display markers related to activation or exhaustion/senescence, along with altered expression of master regulators and of several chemokine receptors. Meanwhile, their CD4+ T cells have an altered cell proliferation but not mitochondria functionality, measured as mitochondrial oxygen consumption and extracellular acidification rate (ECAR). In these patients, a massive cytokine storm detectable in plasma is accompanied by an increased in vitro production of tumor necrosis factor (TNF), interferon (IFN)-γ, interleukin (IL)-2, as well as by a skewing toward the TH17 or Tc17 phenotype. This not only underlines the efficacy of a drug that contrasts cytokine storm, but also suggests the clinical evaluation of novel strategies, based upon the inhibition of the IL-17 pathway.

## Results

**Characteristics of the patients.** We have studied a total of 39 patients, the first 21 of which were consecutively admitted into the Infectious Diseases Clinics of the University Hospital in Modena over the period of March 12th–March 30th, 2020, whereas the remaining patients were studied between April 15th and May 10th. All patients had symptoms including sore throat, fever, cough, dyspnoea, and chest pain. At admission, SARS-CoV-2 was detected in a nasopharyngeal swab specimen by real-time reverse-transcriptase PCR, and in all of them pneumonia was subsequently confirmed by X-ray. Supplementary Tables 1–4 show in details the individual clinical and biohumoral characteristics of the patients. Note that, in general, they were lymphopenic (mean: 1150 cells/μL, range: 360–2334), and in some cases lymphocyte count was very low. For the immunological analyses described below, they were compared with a total of 25 age- and sex-matched healthy controls. For those controls in which we investigated T cell phenotypes, we could measure the lymphocyte absolute number, whose mean was 2320 cells/μL, range: 1580–3130 (controls vs. patients: $p < 0.0001$).

**Characterization and count of CD4+ T cell subsets.** We studied CD4+ and CD8+ T cells in patients and controls using 18 parameter flow cytometry. The analysis of the data was then performed by using two complementary methodologies. In the first, we used the classical approach based upon the two-dimensional recognition of a given cell type, followed by a first gate that was required to identify the population of interest (i.e., CD4+ or CD8+ lymphocytes), followed by sequential gates to identify markers of activation, differentiation, senescence, exhaustion, regulatory CD4+ T cells, and memory stem T cells (TSCM). Data obtained with this gating strategy were then analyzed by appropriate, non-parametric statistical tests, and represented in the figures as scatter plots with means and standard errors of the mean. In the second, we performed an unsupervised analysis that considers the entire, complex scenario presented by CD4+ or CD8+ T cells. This analytical technique uses the multidimensional information obtained by FlowSOM meta-clustering coupled to a dimension-reduction method such as the Uniform Manifold Approximation and Projection (UMAP). Heat maps finally report statistical analysis (see Methods for details). The same approach was used to study CD8+ T lymphocytes.

Figure 1a shows the aforementioned gating strategy that we used to compare CD4+ T cells of healthy donors and COVID-19 patients. A first gate was set on the parameters "time" and "viability", then a second gate was set on CD3+CD45+ cells (i.e., T lymphocytes), among which we identified CD4+ or CD8+ lymphocytes. Within the CD4+ population, we analyzed markers commonly related to T cell activation (HLA-DR and CD38), senescence and exhaustion (CD57 and PD1), differentiation (CD45RA, CCR7, CD28, and CD27), regulatory T cells (Treg, considered as CD25high and CD127low), among which we could identify diverse differentiation stages, and finally TSCM (i.e., CD95+ cells within CD45RA+CCR7+CD27+CD28+ naïve lymphocytes).

Figure 1b shows that patients had a similar percentage of CD4+ T cells to controls, but the absolute number of these cells was significantly lower. A similar phenomenon was observed as far as naïve, central memory, and effector memory CD4+ T cells were concerned, whereas the percentage but not the absolute number of terminally differentiated (TE) cells was higher in patients. Figure 1c reports that patients also expressed higher percentages, but not absolute numbers, of activated cells (co-expressing HLA-DR and CD38), of senescent/exhausted cells (PD1+CD57+) and of regulatory T cells (Treg).

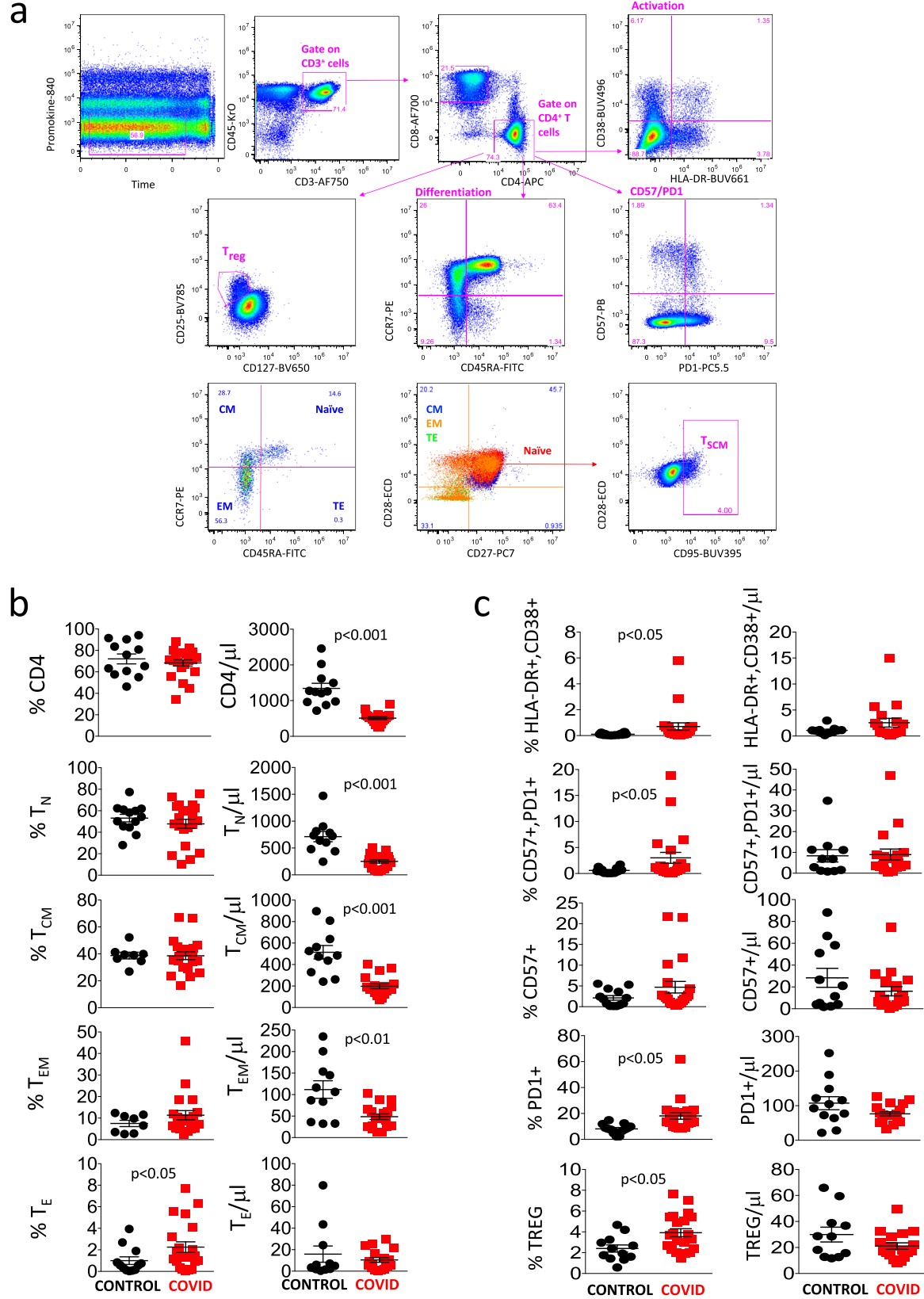

We then used a more sophisticated approach to detect fine changes occurring within different subpopulations of CD4+ T cells. For each patient and control, data from 5000 CD45+ CD3+CD4+ T cells were exported and concatenated in a unique matrix. We explored the T helper cell panel by unsupervised analysis using FlowSOM[14]; this performs multivariate clustering of cells based on the self-organized map (defined "SOM") algorithm, categorizing cells into relevant meta-clusters based on their surface markers. We first clustered all individual cells into 25 distinct clusters based on the surface expression

**Fig. 1 Differentiation, activation, and exhaustion of CD4$^+$ T cells. a** Gating strategy used to analyze markers related to differentiation, activation status, senescence, and exhaustion together with identification of Tregs and TSCM within CD4$^+$ T cells. Naïve T cells are identified as CCR7$^+$CD45RA$^+$CD28$^+$ CD27$^+$ cells; TSCM are CCR7$^+$CD45RA$^+$CD28$^+$CD27$^+$CD95$^+$; central memory (CM) are CCR7$^-$CD45RA$^-$CD28$^+$CD27$^{+/-}$; effector memory (EM) are CCR7$^-$CD45RA$^-$CD28$^{+/-}$CD27$^{+/-}$; terminal effector (TE) are CCR7$^-$CD45RA$^+$CD28$^-$CD27$^{+/-}$. Activated cells are CD38$^+$HLA-DR$^+$; Treg are CD127$^-$CD25$^+$; exhausted/senescent are PD1$^+$CD57$^+$. **b, c** Percentages and absolute numbers of different CD4$^+$ T cell subpopulations in controls ($n =$ 13) and patients ($n = 21$), obtained by manual gating. Data represent individual values, mean (centre bar) ± SEM (upper and lower bars). Statistical analysis by two-sided Mann–Whitney nonparametric test; if not indicated, $p$ value is not significant. Source data are provided as a Source Data file.

marker proteins. Then, to reduce complexity, we merged the clusters that were very close each other and further re-clustered cells into 15 meta-clusters representing different T cell types on the basis of activation, differentiation, and exhaustion. Doing so, we used a dimensionality reduction method, the "UMAP" to distinguish several CD4$^+$ T cell populations (Fig. 2a), whose percentages are reported in the heat map shown in Fig. 2b. It is possible to immediately recognize the high amount of naïve T cells (red dots), that were CD45RA$^+$ CD28$^+$CCR7$^+$ CD27$^+$CD127$^+$CD25$^+$CD95$^-$CD38$^-$HLA-DR$^-$ [15], and that were similar between the two groups; then, we identified recently activated naïve T cells expressing CD38, and those expressing HLA-DR. We also found a small percentage of T cells representing CD4$^+$ memory stem cells characterized by the expression of CD95 and CD38[16], that was similar across the two groups.

Central memory T cells are characterized by expression of CD45RA, CD28, CD27, CD127, and CD95 molecules. Within these, a population expressing only CD38 has been identified, and a population of cells that were activated (HLA-DR$^+$CD38$^+$) and also expressed PD1. In patients, these two populations were significantly more frequent than in controls.

Regarding the effector memory compartment, part of the transitional effector memory T cells are characterized by the lack of expression of CD45RA and CCR7, but express CD28. Effector memory cells were characterized by the lack of expression of CD45RA, CCR7, CD27, CD28, CD38, and HLA-DR and mid expression of PD1, CD127, and CD25; in addition, effector memory expressing marker related to exhaustion (like PD1) were significantly higher in the patients' group. Terminally effector memory T cells that express CD45RA, CD25, CD127, and CD95 did not differ between the two groups, as well as TSCM.

Finally, we are well aware that T regulatory cells (Treg) can be better identified by the expression of Foxp3, which is not present in our panel. However, thanks to the expression of CD127 (low) and CD25 (high), we were able to identify three different putative populations of T regulatory cells: those naïve (expressing CD45RA$^+$ CCR7$^+$), central memory (CD45RA$^-$CCR7$^+$) and the most represented subset of effector memory (CD45RA$^-$CCR7$^-$). The comparison of different clusters between healthy donors and COVID-19 patients is shown in Fig. 2c. It is to note that patients were characterized by higher percentage of naïve Treg cells ($p$ value was lower than the lower limit of the statistical analysis provided the software, i.e., <10$^{-9}$), by more central memory Treg cells, and by increased activated central memory cells expressing PD1.

We were able to analyze in more details cells from seven patients and six controls, measuring the expression of several chemokine receptors and of molecules that are considered master regulators (i.e., lineage-specifying transcription factors). Figure 2d shows a representative comparison of the cytometric analysis within CD3$^+$CD4$^+$ cells in a control donor (upper panels) and a COVID-19 patient. The gating strategy for the identification of CD4$^+$ T cells is reported in Supplementary Fig. 1.

As indicated in Fig. 2e, patients displayed a lower percentage of cells expressing CCR6 or CXCR3, and of those co-expressing CCR6 and CD161, but higher percentages of CXCR4$^+$ or CCR4$^+$ cells; no differences were noted in the expression of T-bet or GATA3.

**Characterization and count of CD8$^+$ T cell subsets**. Figure 3a shows the manual gating strategy that we used for the identification of different types of CD8$^+$ T cells, which was similar to the strategy used for CD4$^+$ T cells. As shown in Fig. 3b, healthy donors and COVID-19 patients displayed a comparable percentage of total CD8$^+$ T cells, even if the absolute number was lower in patients. Both the percentage and absolute number of naïve and central memory cells were lower in patients, while the percentage, but not the absolute number of TE cells was higher.

Figure 3c reports that patients had a higher percentage and absolute number of activated cells (expressing HLA-DR and CD38), and higher percentages of cells expressing PD1 and/or CD57, markers that are associated to cell senescence/exhaustion. No differences were present as far as stem cell memory cells were concerned, similarly with what observed among CD4$^+$ T lymphocytes (not shown).

We then performed the same unsupervised analysis by using FlowSOM meta-clustering to investigate the phenotype of CD8$^+$ T cells, represented by the UMAP in Fig. 4a and by the heat map in Fig. 4b. In this case, 13 clusters were identified and analyzed.

As reported in Fig. 4c, showing the statistical analysis of chemokine receptors and master regulators in patients vs. controls, COVID-19 patients were characterized by higher percentages of terminal effector cells expressing CD38 alone or in combination with CD57, and by activated effector memory cells expressing PD1 or CD57. They also displayed significantly lower percentages of naïve and central memory T cells, which could suggest that patients displayed an exhausted CD8$^+$ T cell compartment.

Figure 4d shows a representative comparison of the cytometric analysis of CD3$^+$CD8$^+$ cells in a control donor (upper panels) and in a COVID-19 patient (lower panels). The gating strategy for the identification of CD8$^+$ T cells is reported in Supplementary Fig. 1. We could also identify CD8$^+$ T lymphocytes expressing high levels of CD161 that, along with CCR6, are present in most mucosal associated invariant T (MAIT) cells characterized by Vα7.2 T cell receptor[17]. Concerning the expression of chemokine receptors and of transcription factors among CD8$^+$ T cells, we found that patients expressed lower percentages of CCR6$^+$, CXCR3$^+$, T-bet$^+$ or CD161$^{high}$ cells, and of CXCR3$^+$ T-bet$^+$ or CCR6$^+$CD161$^+$ lymphocytes (Fig. 4e); they also had higher percentages of cells expressing CCR4, CXCR4, or GATA3.

**T cell proliferation and mitochondrial bioenergetics**. Figure 5a, related to CD4$^+$ cells, shows in the upper panels the gating strategy for the identification of cells that had undergone either cell proliferation (giving origin to the so-called "proliferation index", PI, revealed by those events in the red rectangles) or that replicated different times ("percentage divided", PD, related to the different fluorescence peaks). A representative example of a healthy control and a COVID patient is shown. Lower panels compare five patients and five controls, and show that TE CD4$^+$ T cells from patients had a lower proliferation index. Figure 5b shows the same analysis applied to CD8$^+$ T cells. In this case, patients showed both a significantly lower proliferation index and a higher percentage of dividing cells. The gating strategy for

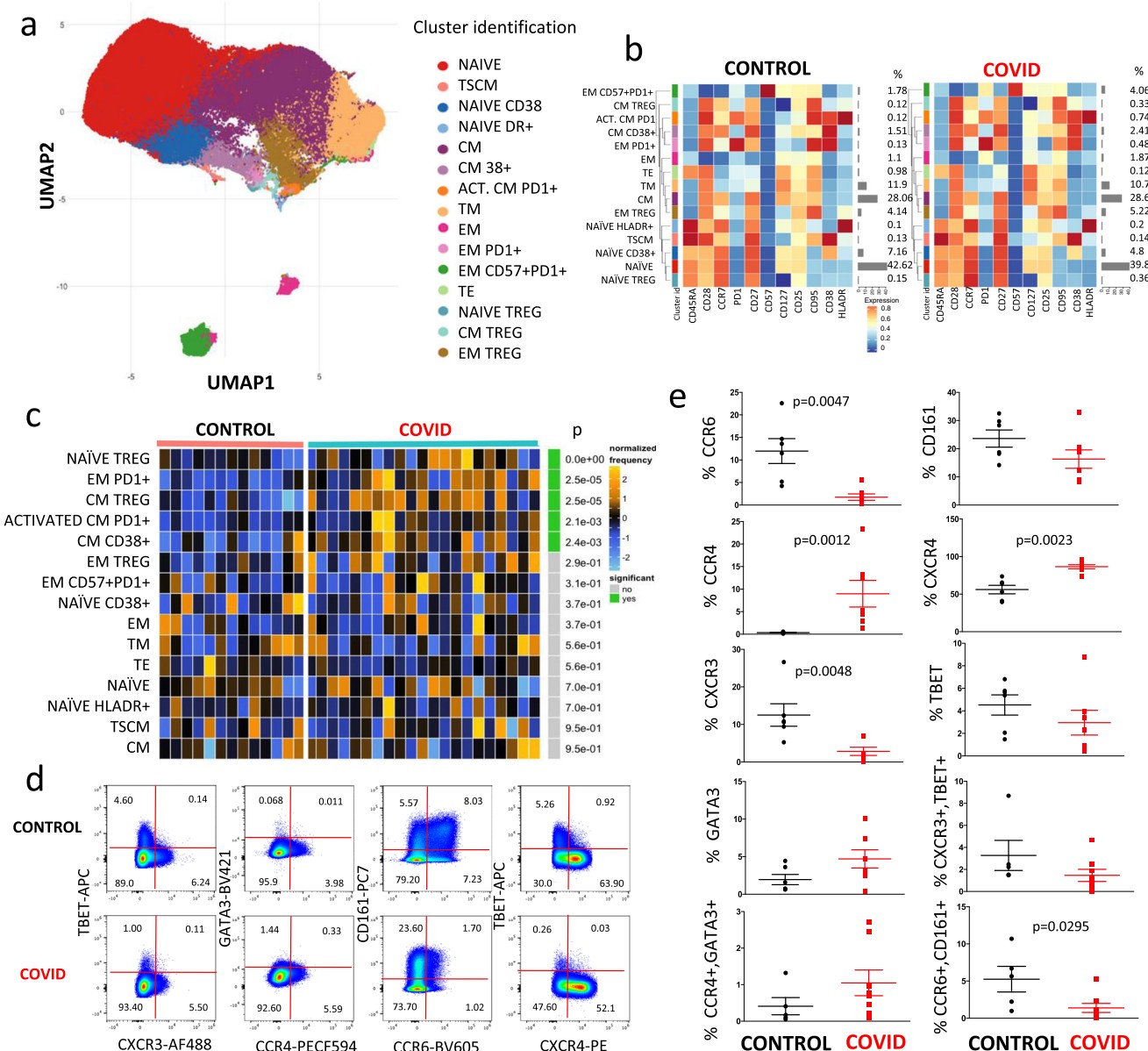

**Fig. 2 Unsupervised analysis of CD4+ T cells and their characterization. a** Uniform Manifold Approximation and Projection (UMAP) representation of the CD4+ T cell landscape. **b** Heat map representing different CD4+ T cell clusters identified by FlowSOM, with relative identity and percentages in healthy controls and COVID-19 patients. The colors in the heat map represent the median of the arcsinh, 0–1 transformed marker expression calculated over cells from all the samples, varying from blue for lower expression to red for higher expression. The dendrogram on the left represents the hierarchical similarity between the metaclusters (metric: Euclidean distance; linkage: average). Each cluster has a unique color assigned (bar on the left). Barplot along the rows (clusters) and values on the right indicate the relative sizes of clusters. **c** Differential analysis between controls (bar color: salmon; $n = 13$) and COVID-19 (emerald; $n = 21$). The heat represents arcsine-square-root transformed cell frequencies that were subsequently normalized per cluster (rows) to mean of zero and standard deviation of one. The color of the heat varies from blue indicating relative under-representation to orange indicating relative over-representation. Bar and numbers at the right indicate significant differentially abundant clusters (green) and adjusted $p$ values. Clusters are sorted according to adjusted $p$ values, so that the cluster at the top shows the most significant abundance changes between the two conditions. **d** Representative dot plots related to the expression of different chemokine receptors and lineage-specifying transcription factors in gated CD4+ T from a control (upper) and a patient (lower panel). Numbers indicate the percentage in each quadrant. Two experiments (one for the control group, one for patients) out of 13 are shown. Numbers indicate the percentage in each quadrant. The gating strategy for the identification of CD4+ T cells is reported in Supplementary Fig. 1. **e** Percentages of different CD4+ T cell subpopulations in controls ($n = 6$) and patients ($n = 7$), obtained by manual gating. Data represent individual values (dots), mean (centre bar) ± SEM (upper and lower bars). Statistical analysis is performed by two-sided Mann–Whitney nonparametric test; if not indicated, $p$ value is not significant. Source data are provided as a Source Data file.

studying cell proliferation in CD4+ and CD8+ T cells is reported in Supplementary Fig. 2.

Cellular bioenergetics was studied in CD4+ T cells purified by magnetic sorting and stimulated overnight or not with anti-CD3 plus anti-CD28 mAbs. Figure 6a, related to the analysis of

mitochondrial oxygen consumption rate (OCR), shows that basal respiration, maximal respiration, spare respiratory capacity (SRC), and the amount of oxygen used for ATP production were similar in stimulated (S) or non-stimulated (NS) cells from controls and patients. Figure 6b, related to the

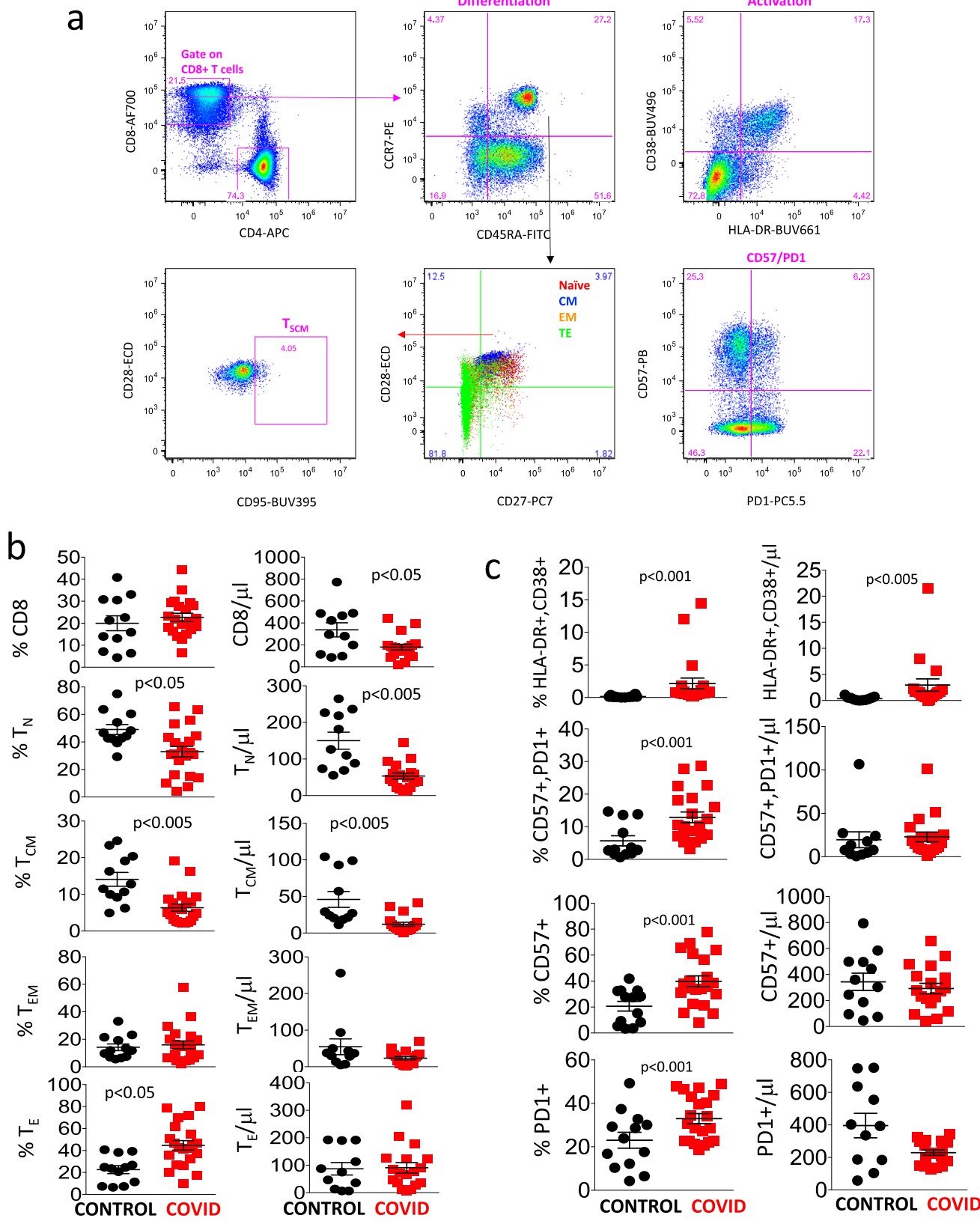

analysis of ECAR, shows that stimulated (S) or NS cells from patients and controls had the same capacity to switch from respiration to glycolysis, and had similar basal and maximal ECAR.

**Detecting cytokine storm in patients' plasma**. We measured plasma levels of 31 cytokines, chemokines, and immune-related molecules in 21 patients and 13 healthy controls (Fig. 7). We found that galectin-1, galectin-3, galectin-9, C-C motif chemokine

**Fig. 3 Differentiation, activation, and exhaustion of CD8+ T-cell subsets. a** Gating strategy used to analyze markers of differentiation, activation status, senescence, and exhaustion, together with identification of TSCM within CD8+ T cells. Naïve T cells are identified as CCR7+CD45RA+CD28+CD27+; TSCM are CCR7+CD45RA+CD28+CD27+CD95+; central memory (CM) are CCR7−CD45RA−CD28+CD27+/−, effector memory (EM) CCR7−CD45RA−CD28+/−CD27+/−; terminal effector (TE) are CCR7−CD45RA+CD28−CD27+/−. Activated cells are CD38+HLA-DR+; exhausted/senescent are PD1+CD57+. **b, c** Percentages and absolute numbers of different CD8+ T cell subpopulations in controls (n = 13) and patients (n = 21), obtained by manual gating. Data represent individual values, mean (centre bar) ± SEM (upper and lower bars). Statistical analysis by two-sided Mann–Whitney nonparametric test; if not indicated, p value is not significant. Source data are provided as a Source Data file.

ligand-2 (CCL2, also known as MCP-1), CCL3 (also known as MIP-1a), CCL4 (also known as MIP-1b), C-X-C motif chemokine ligand-6 (CXCL6), CD27, CD40, CD40L, TNF, IFN-γ, IL-1α, IL-1β, IL-4, IL-6, IL-7, IL-8, IL-10, IL-13 major histocompatibility complex class I chain-related protein A (MICA), glucocorticoid-induced TNFR-related protein (GITR, also known as receptor superfamily member 18), and programmed death-ligand 1 (PD-L1) were significantly higher in COVID-19 patients, as compared with controls. No statistical differences were found as far as MIP-2 (also known as CXCL2) was concerned.

Likely due to technical limitations of the multiplex assay, CCL7, GITR, IL-2, IL-3, and IL-5 were not detected in most patients and controls. Also due to the fact that levels of IL-12p70, IL-15, and IL-17 were undetectable in almost all of the healthy controls, we were unable to perform statistical comparisons, even although in most patients these cytokines were present at detectable levels.

**In vitro production of proinflammatory cytokines**. We studied six different functions of CD4+ and CD8+ T cells, including the production of IFN-γ, IL-17, IL-2, TNF and granzyme-B, along with the expression of the degranulation marker CD107a[18].

Figure 8 shows a representative example of the detection of intracellular cytokines and the presence of CD107a in CD4+ T cells (Fig. 8a) or CD8+ T cell (Fig. 8b) from a healthy donor (CONTROL, upper quadrants) and a patient (lower quadrants). The gating strategy for studying intracellular cytokines in CD4+ and CD8+ T cells is reported in Supplementary Fig. 3. As summarized in (Fig. 8c), in comparison with six controls, a significantly higher amount of CD4+ T cells from eight COVID-19 patients was able to produce TNF, CD107a, IFN-γ, IL-2, and particularly IL-17 (that showed the highest difference). The production of granzyme-B was similar in the two groups. A similar trend was present among CD8+ T cells (Fig. 8d), whose stimulation with anti-CD3/CD28 resulted in significantly higher production of CD107a, IL-17, and especially IL-2. A trend, even if not reaching a statistical significance, was present as far as the production of TNF and IFN-γ is concerned.

We then investigated CD4+ and CD8+ T cell polyfunctionality by analyzing the simultaneous production of TNF, CD107a, IFN-γ, IL-2 and IL-17 (Fig. 9). It is to note that in this case we indicate the percentage of cells exerting one or more functions (considering the entire set of cells, that is taken as 100%), without considering the power of the global response[19]. Based on the use of five markers, we could discriminate 31 different populations of CD4+ or CD8+ T cells able to produce one or more molecules; the population that had no activity, i.e., that was five times negative for all five markers, was excluded from the analysis. Figure 9a shows that CD4+ and CD8+ T cells from patients and controls had very global patterns of polyfunctionality. However, different functional types of CD4+ T cells producing TNF were present at higher proportions in patients as compared with controls (Fig. 9b). Moreover, patients also showed more cells simultaneously producing IL-2 and IL-17 than controls, confirming what was reported above.

## Discussion

In this study we describe the main changes in the T cell compartment of patients with COVID-19 pneumonia. Most were lymphopenic, and most did not require noninvasive ventilation, indicating that at the time of blood collection the disease was severe, but not too advanced to require intubation and mechanical ventilation. Our data thus indicate that several immune alterations are present when the infection becomes clinically relevant. These data are also in agreement with the immunological changes recently described in a patient with mild-to-moderate COVID-19 patient that required hospitalization, and showed a significant increase in activated T cells[20]. Studies on asymptomatic, infected individuals are crucial to better understand the immunopathogenesis of COVID-19.

In COVID-19 patients with pneumonia, we found an increase capability of CD4+ or CD8+ T cells to produce in vitro IL-17, that is able to strengthen the inflammation response and to activate neutrophils. In peripheral blood, patients also showed low percentages of both CD4+ and CD8+ T cells expressing CCR6 and high levels of CD161, which is typical of TH17 and of mucosal associated invariant T (MAIT) cells, respectively. The loss of circulating CD4+CD161+CCR6+ cells contributes to disease progression in macaques infected with the simian immunodeficiency virus, and it has been shown that these cells accumulate in the rectal mucosa, enhancing inflammation[21]. In the same infection model, the percentage of IL-17 producing CD8+CD161+ cells present in the lung can be fourfold higher than that in peripheral blood[22]. Furthermore, cells present in the lung were able to produce more IL-17 than those from peripheral blood. Taken together, these findings underline the importance of IL-17 in COVID-19, and likely could pave the way to novel therapeutic approaches based upon IL-17 blockage by biological drugs that are already available.

Very few data exist on the changes in the T cell compartment in patients affected by SARS-CoV-2 infection. Previous studies have shown changes in the T cell family, which is characterized by signs of exhaustion[23,24]. In most COVID-19 patients the proportions of T cells subsets can remain within the normal range, but a decrease can exist in CD4+ and CD8+ T cell counts or in CD4+/CD8+ ratio[25,26]. Our data are in agreement with these observations, but are enhanced by our analyses of different subpopulations of both CD4+ and CD8+ T lymphocytes. Indeed, whereas on the one hand no gross changes between patients and controls are detected simply using markers related to naïve, memory or effector cells, on the other hand our more sophisticated and detailed analysis reveals significant differences.

Treg are crucial for regulating immune homeostasis and self-tolerance[27,28]. They express the forkhead box transcription factor Foxp3, but can also be identified by detecting the high expression of the IL-2 receptor α-chain (CD25) and low/null expression of IL-7 receptor α-chain (CD127)[29], along with other surface molecules such as CD39 and CD73[30,31]. They suppress auto-immune phenomena, dampen allergic reactions, or block transplant rejection, but they can also inhibit a protective immune

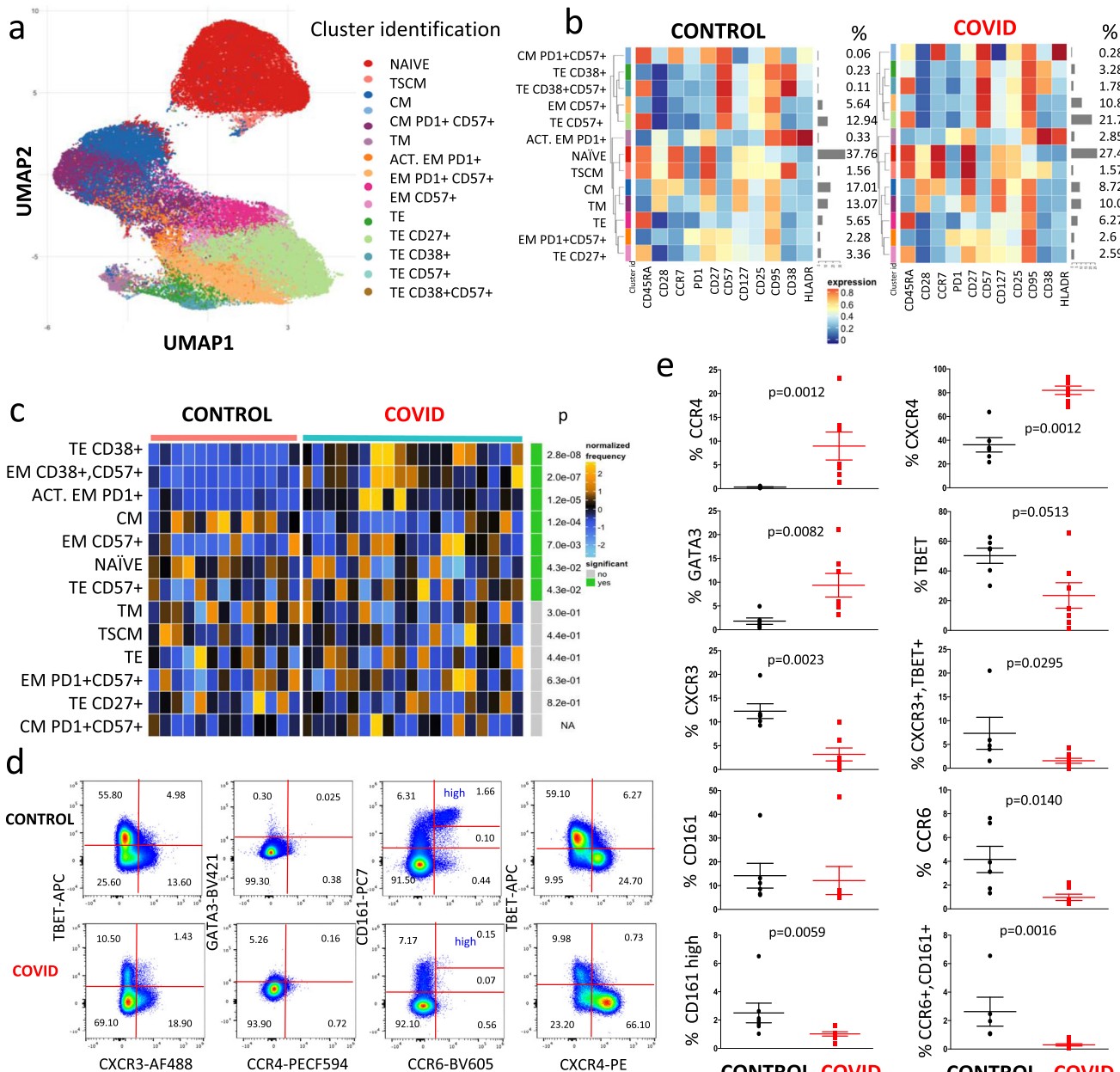

**Fig. 4 Unsupervised analysis of CD8+ T cells and their characterization. a** Uniform Manifold Approximation and Projection (UMAP) UMAP representation of CD8+ T cell landscape. **b** Heat map representing different clusters identified by FlowSOM, with relative identity and percentages in controls and patients. The color in the heat map represents the median of the arcsinh, 0–1 transformed marker expression calculated over cells from all the samples, varying from blue for lower expression to red for higher expression. The dendrogram on the left represents the hierarchical similarity between the metaclusters (metric: Euclidean distance; linkage: average). Each cluster has a unique color assigned (bar on the left). Barplots along the rows (clusters) and values on the right indicate the relative sizes of clusters. **c** Differential analysis between CTR (bar color: salmon; $n = 13$) and COVID-19 (emerald; $n = 19$). The heat represents arcsine-square-root transformed cell frequencies that were subsequently normalized per cluster (rows) to mean of zero and standard deviation of one. The color of the heat varies from blue indicating relative under-representation to orange indicating relative over-representation. Bar and numbers at the right indicate significant differentially abundant clusters (green) and adjusted $p$ values. Clusters are sorted according to adjusted $p$ values, so that the cluster at the top shows the most significant abundance changes between the two conditions. **d** Representative dot plots related to the expression of different chemokine receptors and lineage-specifying transcription factors in gated CD8+ T from a control donor (upper) and a patient (lower panel). Two experiments (one for the control group, one for patients) out of 13 are shown. Numbers indicate the percentage in each quadrant. The gating strategy for the identification of CD8+ T cells is reported in Supplementary Fig. 1. **e** Percentages of different CD8+ T cell subpopulations in controls ($n = 6$) and patients ($n = 7$), obtained by manual gating. Data represent individual values (dots), mean (centre bar) ± SEM (upper and lower bars). Statistical analysis is performed by two-sided Mann–Whitney nonparametric test; if not indicated, $p$ value is not significant. Source data are provided as a Source Data file.

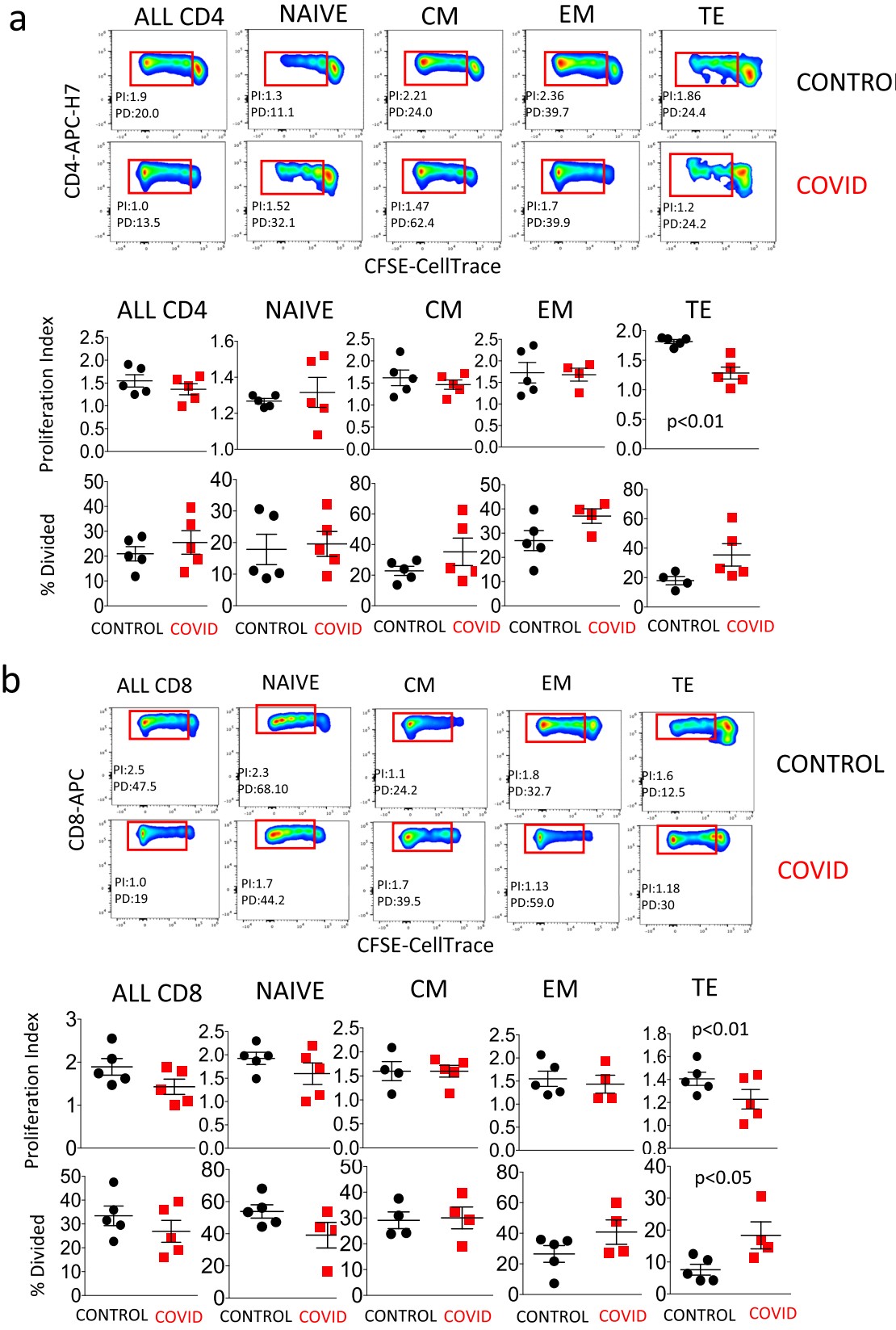

response against invading pathogens or tumors[32]. Here we show that the percentages of different types of Tregs are increased in peripheral blood from COVID-19 patients, and that their plasma contains high amounts of the inhibitory cytokine IL-10.

Patients with COVID-19 had increased amounts of CD8$^+$ T cells expressing CD57, which is considered a key marker of in vitro replicative senescence and is associated either to human aging or prolonged chronic infections[33]. CD57 has been used to

**Fig. 5 Cell proliferation of CD4+ and CD8+ T lymphocytes. a** Upper two rows: representative dot plots related to cell proliferation in different types of CD4+ T cells from a control donor (upper panel) and a patient (COVID, lower panel). Lower two rows indicate the proliferation index and the percentage of divided cells in all CD4+ T cells, or in naive, central memory (CM), effector memory (EM), or terminally differentiated (TE) cells. Data represent individual values (dots) from five patients and five controls, mean (centre bar) ± SEM (upper and lower bars). Statistical analysis is performed by two-sided Mann–Whitney nonparametric test; if not indicated, *p* value is not significant. The gating strategy for the identification of CD4+ T cells is reported in Supplementary Fig. 2. **b** Upper two rows: representative dot plots related to cell proliferation in different types of CD8+ T cells from a control donor (CTR, upper) and a patient (COVID, lower panel). Lower two rows indicate the proliferation index and the percentage of divided cells in all CD8+ T cells, or in naive, central memory (CM), effector memory (EM), or terminally differentiated (TE) cells. Data represent individual values (dots) from five patients and five controls, mean (centre bar) ± SEM (upper and lower bars). Statistical analysis is performed by two-sided Mann–Whitney nonparametric test; if not indicated, *p* value is not significant. The gating strategy for the identification of CD8+ T cells is reported in Supplementary Fig. 2.

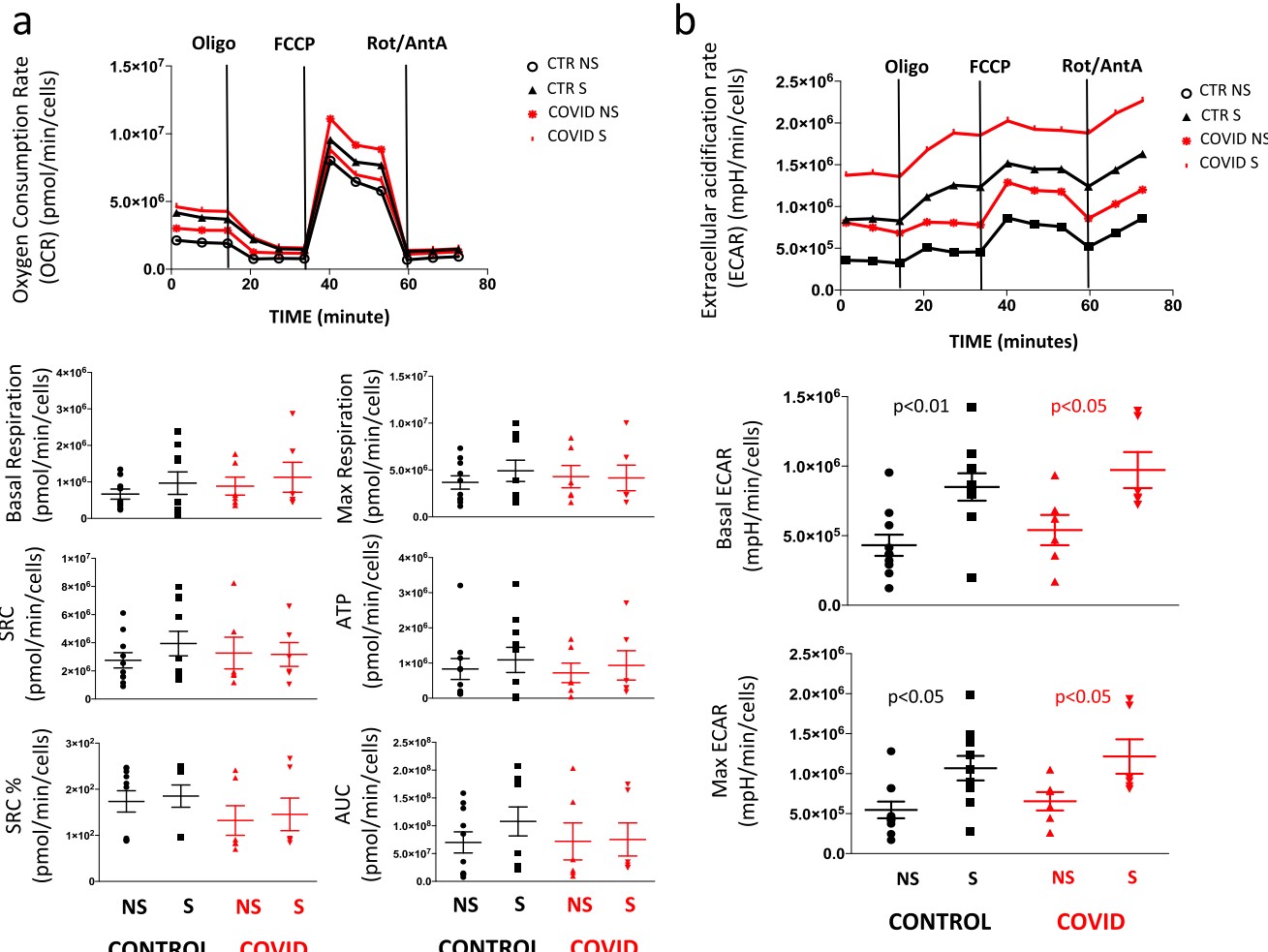

**Fig. 6 Mitochondria bioenergetics of CD4+ T cells. a** Representative traces (out of 12 experiments) of oxygen consumption rate (OCR) of unstimulated (NS) and stimulated (S) CD4+ T cells from healthy controls (CTR; $n = 7$) and COVID patients ($n = 5$). OCR was measured in real time, under basal condition and in response to indicated mitochondria inhibitors: oligomycin (Oligo, 2 μM), cyanide-4-(trifluoromethoxy)phenylhydrazone (FCCP, 0.5 μM), and antimycin A plus rotenone (Rot/AA, 0.5 μM). Histograms show the quantification of basal respiration, maximal respiration, spare respiratory capacity (SRC), ATP-linked respiration (ATP), percentage of SRC, and area under the curve (AUC) in cell stimulated (S) with anti-CD3/28 or non-stimulated (NS) from controls and patients. The percentage of SRC represents the ratio between the seventh and the third measurement. AUC was obtained by analyzing the area under the curve from the sixth to the eleventh measurement. Data represent individual values (dots) from seven patients and five controls, mean (centre bar) ± SEM (upper and lower bars). Statistical analysis by two-sided Mann–Whitney nonparametric test shows no statistical differences between controls and patients. **b** Representative traces (out of 12 experiments) of extracellular acidification rates (ECAR) of unstimulated (NS) and stimulated (S) CD4+ T cells from one healthy control (CTR) and one COVID patient. ECAR was measured under basal conditions and in response to FCCP. Data represent individual values from seven controls and five patients, mean (centre bar) ± SEM (upper and lower bars) concerning the quantification of basal ECAR and maximal ECAR in cell stimulated (S) with anti-CD3/28 or non-stimulated (NS) from controls and patients. Statistical analysis by two-sided Mann–Whitney nonparametric test shows no statistical differences between controls and patients.

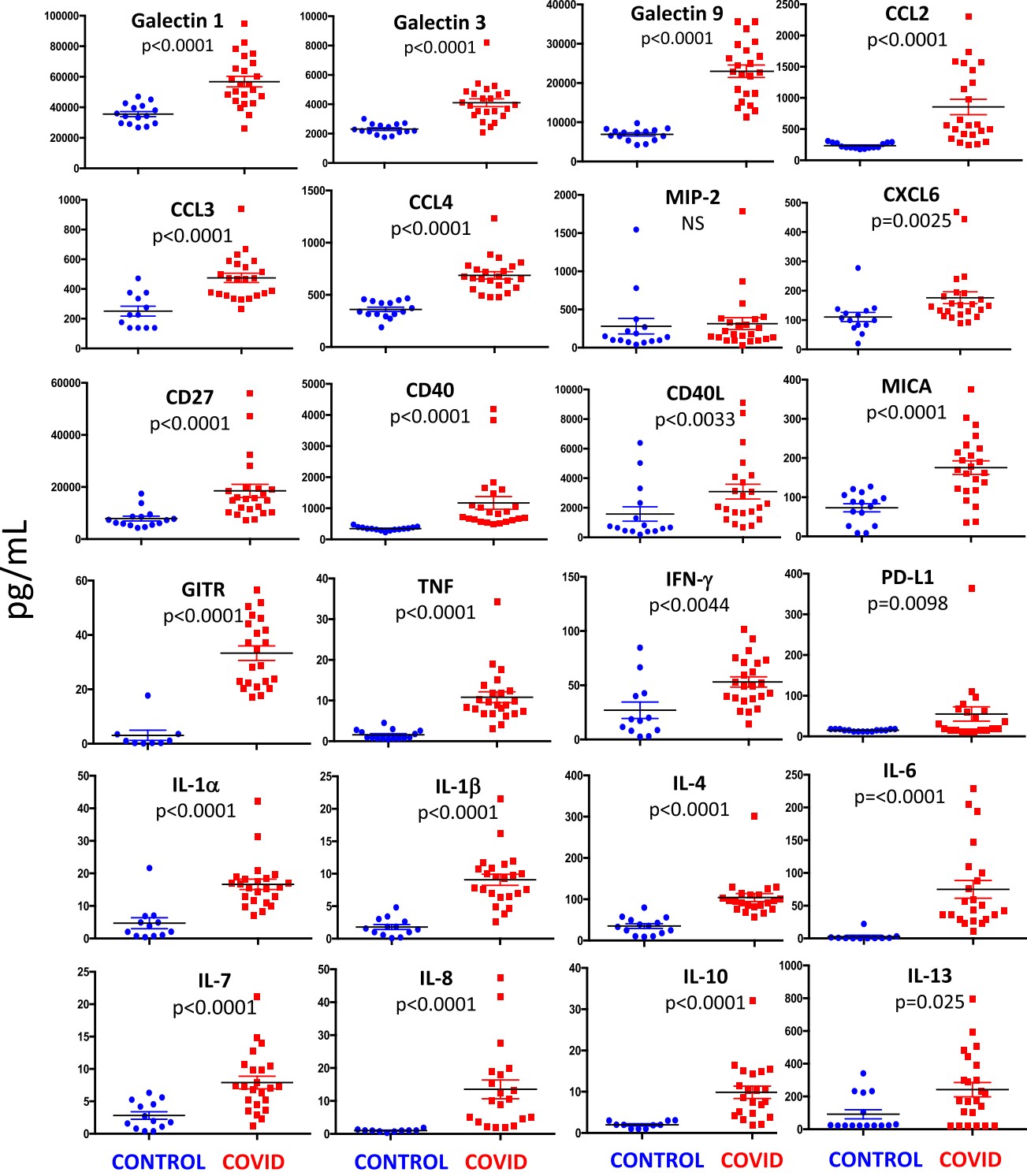

**Fig. 7 Plasma level of cytokines and chemokines from COVID-19 patients and controls.** Quantification of cytokines and other mediators in plasma obtained from COVID-19 patients ($n = 23$) and healthy controls ($n = 15$). Data represent individual values, mean (centre bar) ± SEM (upper and lower bars). Statistical analysis by two-sided Mann–Whitney nonparametric test; if not indicated, $p$ value is not significant. Source data are provided as a Source Data file.

detect functional immune deficiency in patients with auto-immune diseases, infectious diseases, and cancers. It has been suggested that CD57[+] cells display a high susceptibility to activation-induced cell death, and are not able to undergo cell proliferation despite their preserved ability to secrete cytokines after activation[34]. We have found a decreased proliferation index among TE CD4[+] and CD8[+] T lymphocytes from COVID-19 patients. T cells show phenotypic features of an exhausted state including the upregulated expression of the inhibitory receptors such as PD1. T cell exhaustion is

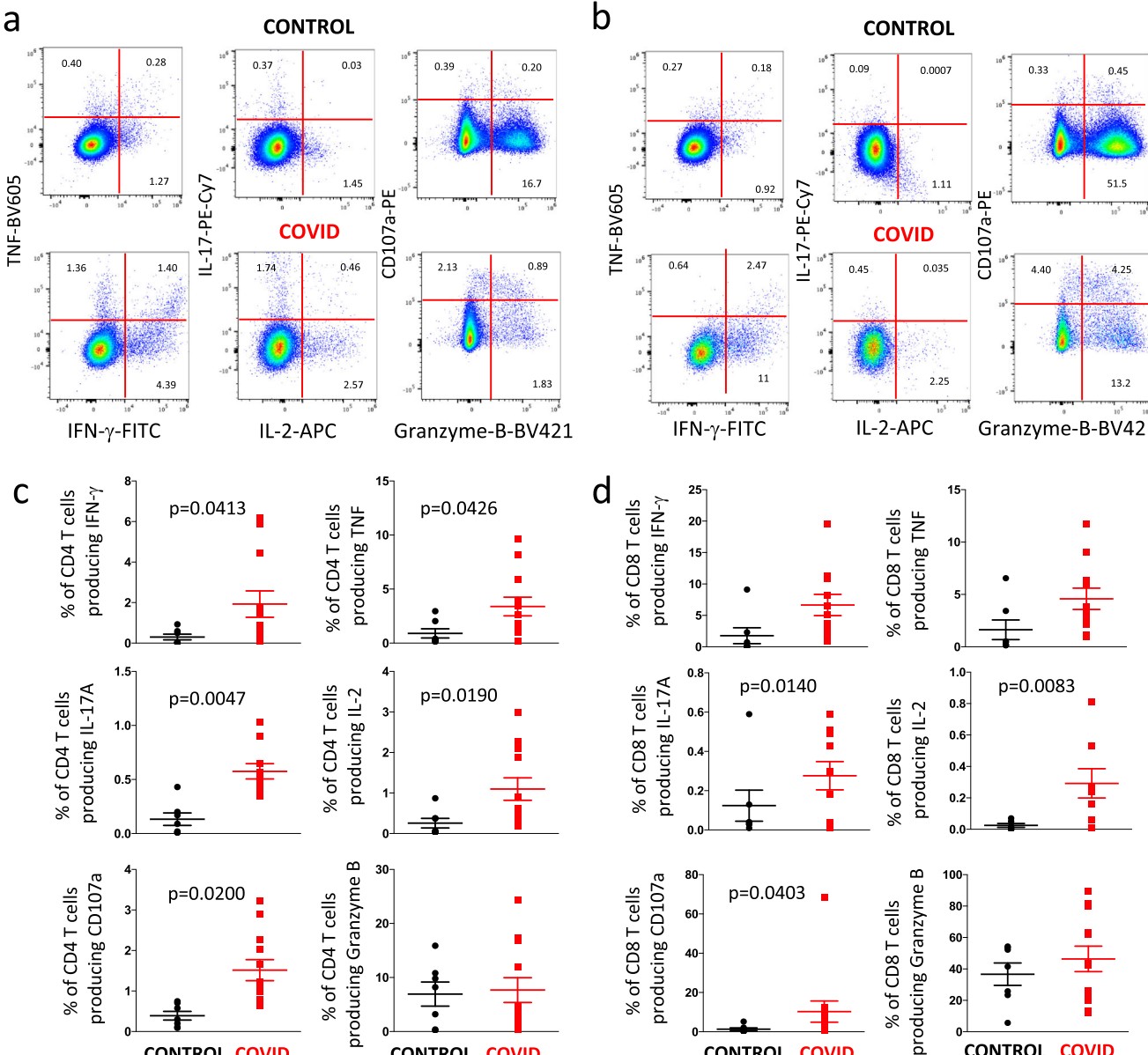

**Fig. 8 Cytokine production by CD4$^+$ and CD8$^+$ T cells after in vitro stimulation with anti-CD3/28. a** Representative dot plots (out of 18 experiments) related to the intracellular cytokine staining of CD4$^+$ T cells of a control donor (upper panels) and a COVID patient (lower panels) after in vitro stimulation with anti-CD3/CD28. The gating strategy for the identification of CD4$^+$ T cells is reported in Supplementary Fig. 3. **b** Representative dot plots (out of 18 experiments) related to the intracellular cytokine staining of CD8$^+$ T cells of a control donor (upper panels) and a COVID patient (lower panels) after in vitro stimulation with anti-CD3/CD28. The gating strategy for the identification of CD8$^+$ T cells is reported in Supplementary Fig. 3. **c** Comparison between the total production of IFN-γ, TNF, IL-17, IL-2, CD107a and granzyme-B by CD4$^+$ T cells after in vitro stimulation with anti-CD3/CD28. Data represent individual values from six controls and eight patients, mean (centre bar) ± SEM (upper and lower bars). Statistical analysis by two-sided Mann–Whitney nonparametric test; if not indicated, *p* value is not significant. **d** Comparison between the total production of IFN-γ, TNF, IL-17, IL-2, CD107a and granzyme-B by CD8$^+$ T cells after in vitro stimulation with anti-CD3/CD28. Data represent individual values from six controls and eight patients, mean (centre bar) ± SEM (upper and lower bars). Statistical analysis by two-sided Mann–Whitney nonparametric test; if not indicated, *p* value is not significant.

characterized by functional unresponsiveness, which prevents massive immunoactivation and associated autoimmune tissue damage. Thus, it could be possible that in COVID-19 patients the activation of these cells is followed not only by the lack of clonal expansion (as revealed by the decreased proliferation) but also by the production of molecules that cause inflammation.

Furthermore, we have observed that cells from patients displayed higher percentages of different cell types expressing of

PD1, a crucial molecule for the induction and maintenance of peripheral tolerance, and for maintaining T cells stability and integrity[35]. The PD1/PD-L1 axis also mediates potent inhibitory signals to block proliferation and function of T effector cells, causing inimical effects on antiviral immunity. We have found significantly high plasma levels of PD-L1 in our patients, and studies are needed to understand whether and how the infection uses this pathway for inhibition of an efficient antiviral immune response.

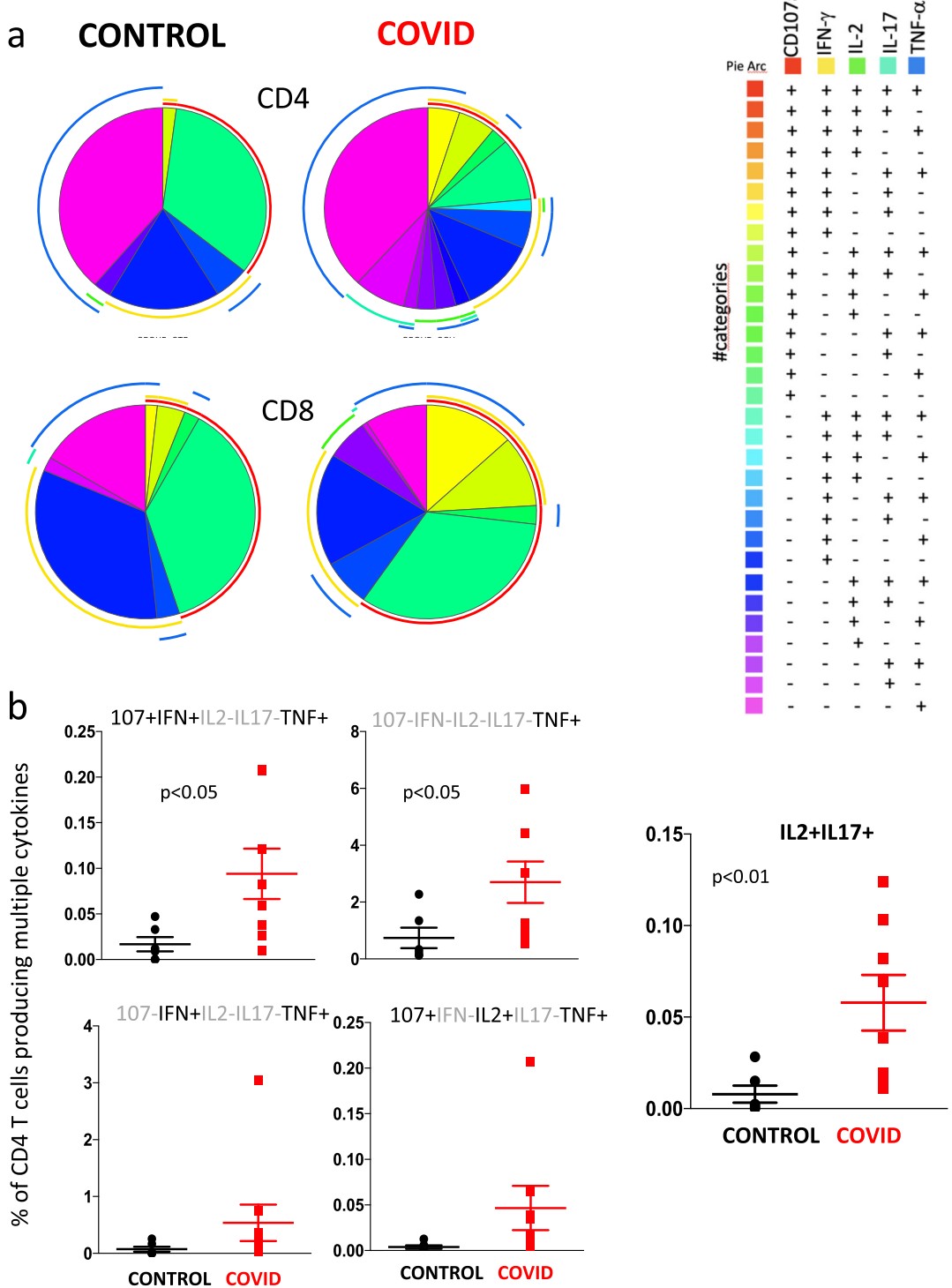

**Fig. 9 Polyfunctionality of CD4$^+$ and CD8$^+$ T cells after in vitro stimulation with anti-CD3/28. a** Pie charts representing the proportion of responding CD4$^+$ (upper pies) and CD8$^+$ (lower pies) T cells producing different combinations of CD107a, IL-2, IFN-γ, and TNF after anti-CD3/CD28 stimulation in control donors (left; $n = 7$) and patients (right; $n = 11$). Frequencies were corrected by background subtraction as determined in non-stimulated controls using SPICE software. Pie arches represent the total production of different cytokines. **b** Comparison between the production of different combinations of cytokines by CD4$^+$ T cells after in vitro stimulation with anti-CD3/CD28. Data represent individual values from 7 controls and 11 patients, mean (centre bar) ± SEM (upper and lower bars). Statistical analysis by two-sided Mann–Whitney nonparametric test; if not indicated, p value is not significant.

An excessive inflammatory response evidenced by elevated levels of proinflammatory cytokines and chemokines has been described in patients affected by the SARS-CoV epidemic in 2003[36,37]. Moreover, in vitro experiments revealed that several different cell types from those patients were able to produce high amounts of cytokines[38,39]. High plasma levels of proinflammatory molecules indicating a TH1/TH17 response were also reported in patients with MERS-CoV infection, with increased concentrations

of IFN-γ, TNF, IL-15, and IL-17[40]. Similar data, along with skewed in vitro cytokine production by T cells, have been described in patients infected by MERS coronavirus[41]. In our COVID-19 patients, plasma concentrations of many proinflammatory cytokines and chemokines were dramatically increased, which is in accordance to all the aforementioned reports describing the so-called "cytokine storm"[42].

The effects of the presence of high amounts of molecules that are produced by several cell types and act on innate immune cells must be considered. Amongst these molecules, IL-8 might be of a particular importance, since it recruits neutrophils from the blood to infected or injured tissue. IL-8 production can be induced by a wide range of stimuli, including TNF, IL-1, bacteria, viruses, and cellular stress, and it can be synthesized by several cell types, including monocytes, macrophages, endothelial and epithelial cells, fibroblasts, T lymphocytes, hepatocytes, synovial cells, and keratinocytes[43]. Its receptors, CXCR1 and CXCR2, are expressed on neutrophils, monocytes, CD8$^+$ T and NK cells, mast cells, basophils, and myeloid-derived suppressor cells. In neutrophils, receptor activation stimulates degranulation and the production of reactive oxygen species[44]. When a respiratory virus, such as SARS-CoV-2, enters alveoli, it first encounters and infects alveolar epithelial cells, which can then produce IL-8 that in turn attracts and activates neutrophils and macrophages. These start damaging the organ, eventually triggering a much more complex series of pathogenic events, including, amongst others, endothelial damage, platelet activation, and intravascular thrombosis. Although it was not possible to precisely determine the site of IL-8 production, even for our patients with pneumonia, it is reasonable to hypothesize that alveolar cells are intimately involved in this phenomenon.

Surprisingly, we also observed a marked plasma increase of typical TH2 cytokines, including IL-4, IL-10, and IL-13. This could indicate that the activation of immune system is really massive, indiscriminately involving all cells, and, similarly to what occurs during bacterial sepsis, could cause a form of immune paralysis[45].

Galectin-1, galectin-3, and galectin-9 were increased in patients as compared with controls. Galectins represent a family of soluble β-galactoside-binding proteins widely expressed at sites of inflammation and infection that have emerged as a new class of damage-associated molecular patterns (DAMPs) or resolution-associated molecular patterns, serving to magnify or to otherwise correct inflammatory responses[46,47]. In particular, galectin-1 acts typically as a proresolving mediator by repressing a number of innate and adaptive immune programs. Conversely, galectin-3 and -9 have been proposed to act as alarmins (or DAMPs) that amplify inflammatory responses during sepsis and several types of infection. To our knowledge, these are the first data describing the presence of these types of soluble mediators in patients with COVID-19, and further studies are needed to investigate their importance in the immunopathogenesis of the disease, or their possible consideration as potential therapeutic targets.

The massive production and release of cytokines is very similar to what occurs during polyclonal, superantigen-driven, T-cell activation[48]. A fourfold increase of the levels of a TH1 molecule such as IFN-γ was indeed observed in plasma from COVID-19 patients as compared with controls. IFN-γ activates macrophages, which produce proinflammatory cytokines, which then overwhelm the system. When cells from patients were stimulated with anti-CD3/28, an increased number of CD4$^+$ T cells producing IFN-γ, TNF, IL-17, and IL-2 was observed as compared with healthy controls. This indicates that T cells from COVID-19 patients have a higher ability over controls to respond in vitro to stimulatory challenges producing potentially dangerous molecules. Studies are underway to investigate the functional phenotype of specific T cells, i.e., those responding to peptide antigens derived from SARS-CoV-2.

Elevated levels of serum proinflammatory cytokines and chemokines are known to contribute, as a cytokine storm, to the increased severity of disease caused by some strains of corona virus. Unique immunoregulatory system mediated by T cell exhaustion and suppressive cytokines such as IL-10 are responsible for limiting excessive inflammation and play an important role in homeostasis in the lungs. A balance in the levels of immunoactivation and immunosuppression may therefore be crucial in host defense against highly pathogenic corona virus infection.

We are well aware of the limitations of our study. First of all, the fact that the relatively low number of patients, who were however chosen on the basis of similar clinical characteristics, does not allow us to divide them into those with a mild or severe course of the infection, having sufficient statistical power for further analyses. Second, due to fact that the study started when there was no treatment at all, and that in the days following blood collection some patients would have been treated and others not, it is difficult to understand which immune modifications can be considered predictive markers of the natural containment of the infection, or of the success of the therapy. Third, for this study we could not provide longitudinal data, but just compared the cohort with healthy donors. However, in COVID-19 patients with pneumonia, we have found the presence of: (i) increased markers of T cell exhaustions, activation, and senescence; (ii) an altered differentiation of different T cell subtypes; (iii) high plasma levels of a variety of cytokines, from those with proinflammatory action to those that are able to inhibit the immune response, from those that indicate a skewing towards TH1 to those that reveal a skewing towards TH2; (iv) massive in vitro production of several cytokines, with a potential skewing of activated cells towards TH17 phenotype; and (v) decrease of those circulating cells that are known to localize in the lung and other tissues, and recruit neutrophils.

In conclusion, in COVID-19 patients, concomitant aspects of immune inhibition, activation, exhaustion, and complex alterations within cells at different stage of differentiation exist, and a huge variety of cytokines are produced and released. The overall picture that emerges underlines the ability of SARS-CoV-2 to provoke, in a very short period of time but, fortunately, not in all infected individuals, a massive activation of immune responses. However, and thanks to the considerable efforts that the scientific community is devoting to studies of the immunopathogenesis of COVID-19, the progression of the viral infection starts to show some weak points that may be therapeutically relevant. For example, a therapeutic approach now exists based on the administration of drugs that block IL-6 pathway, and is now significantly ameliorating the course of the disease[13,49]. IL-17 is crucial in recruiting and activating neutrophils, cells that can migrate to the lung and are heavily involved in the pathogenesis of COVID-19. We show here that a significant skewing of T cell activation towards TH17 functional phenotype exists in COVID-19 patients, and we therefore suggest that blocking IL-17 pathway by biological drugs that are already available and used to treat different pathologies could be a novel, additional strategy to treat patients infected by SARS-CoV-2.

## Methods

**Study design.** This is a case-control, cross sectional, single-centre study, approved by the local Ethical Committee (Comitato Etico dell'Area Vasta Emilia Nord, protocol number 177/2020, March 11th, 2020) and by the University Hospital Committee (Direzione Sanitaria dell'Azienda Ospedaliero-Universitaria di Modena, protocol number 7531, March 11th, 2020). Each participant, including healthy controls, provided informed consent according to Helsinki Declaration, and all uses of human material have been approved by the same Committees. A total of 39

COVID-19 patients were included in the study; they had a median age of 64 years (range 35–94), 7 were females, 32 males. Patients were matched for age and gender with a total of 25 healthy donors (CTR), median age 60 years (range 33–66 years). We recorded demographic data, medical history, symptoms, signs, temperature, and main laboratory findings from each patient. Data were coded and recorded in Excel 14.1.0 for Mac in a database present in the Infectious Disease Clinics and routinely used. The total number and type of leukocytes in peripheral blood was analyzed by hemocytometer in the Clinical Laboratory of the University Hospital, according to routine methods.

**Blood collection and isolation of mononuclear cells**. Up to 20 mL of blood were collected from each patient in vacuettes containing ethylenediamine-tetraacetic acid. Blood was immediately processed. Isolation of peripheral blood mononuclear cells (PBMC) was performed using ficoll-hypaque according to standard procedures[50]. PBMC were then stored in different aliquots at the concentration of 5–10 millions/mL in liquid nitrogen in fetal bovine serum supplemented with 10% dimethyl sulfoxide. Plasma was then collected, centrifuged twice, and stored at −80 °C until use. Measurements were taken from individual patients; in the case of plasma, each measurement was performed in duplicate and only the mean was considered and shown.

**T cell immunophenotype by polychromatic flow cytometry**. Thawed PBMC were washed twice with RPMI 1640 supplemented with 10% fetal bovine serum and 1% each of l-glutamine, sodium pyruvate, nonessential amino acids, antibiotics, 0.1 M HEPES, 55 μM β-mercaptoethanol and 0.02 mg/ml DNAse[51].

For the detailed analysis of T cell phenotype, PBMC were counted and up to 1 million PBMC were stained with the Duraclone IM T cell panel (from Beckman Coulter, Miami, FL) added with another five fluorescent mAbs and a marker of cell viability. Along with side and forward scatter signals, signals were obtained from different fluorochrome-labeled mAbs, i.e., CD45 conjugated with Krome Orange, CD3 APC-A750, CD4 APC, CD8 AF700, CD27 PC7, CD57 Pacific Blue, CD279 (PD1) PC5.5, CD28 ECD, CCR7 PE, CD45RA FITC, HLA-DR BUV661, CD127 BV650, CD25 BV785, CD95 BUV395, CD38 BUV496, and PromoFluor-840 (Promokine, PromoCell, Heidelberg, Germany). A minimum of 500,000 cells per sample were acquired on a CytoFLEX LX flow cytometer (Beckman Coulter)[18].

For the analysis of T cell skewing toward TH1, TH2, or TH17, and chemokine receptor expression, thawed PBMC were washed twice with PBS and stained with the viability marker AQUA LIVE DEAD (ThermoFisher). Then, up to 1 million cells were washed and stained at 37 °C with the following mAbs: anti-CXCR3-AF488, -CXCR4-PE. Cells were washed again and stained at room temperature with anti-CD161-PC7, -CCR6-BV605, -CCR4-PE-CF594, -CD4-AF700, -CD8-APC-Cy7. Cells were washed, fixed, and permeabilized using Foxp3/Transcription Factor Staining Buffer Set (ThermoFisher). Finally, cells were stained with anti-GATA3-BV421 and anti-TBET-APC, and washed. A minimum of 500,000 PBMC were acquired by using Attune NxT acoustic focusing flow cytometer (ThermoFisher).

Supplementary Table 5 shows all the clones and monoclonal antibodies used in this study.

**Representation of high parameter flow cytometry**. Flow Cytometry Standard (FCS) 3.0 files were imported into FlowJo software version 9 (Becton Dickinson, San Josè, CA), and analyzed by standard gating to eliminate aggregates and dead cells, and to identify CD3+ CD4+ T cells and CD8+ T cells. Then, data from 5000 CD4+ T cells and 2500 CD8+ T cells per sample were exported for further analysis in R, by following a script that makes use of Bioconductor libraries and R statistical packages (CATALYST 1.10.1). The script is available at: https://github.com/HelenaLC/CATALYST[52]. The selection of cofactor for data transformation was checked on Cytobank premium version (see: cytobank.org). The platform Flow-SOM (available at: https://bioconductor.org/packages/release/bioc/html/FlowSOM.html) was used to perform the metaclustering ($K = 20$). Data were subsequently displayed using the dimensionality reduction method named UMAP[2].

**Quantification of cytokine plasma levels**. The plasma levels of 31 molecular species was quantified using a Luminex platform (Human Cytokine Discovery, R&D System, Minneapolis, MN) for the simultaneous detection of the following molecules: IL-1α, IL-1β, IL-2, IL-3, IL-4, IL-5, IL-6, IL-7, IL-8, IL-10, IL-12p70, IL-13, IL-15, IL-17, galectin-1, galectin-3, galectin-9, IFN-γ, TNF, GITR, PD-L1, MICA, CCL-2, CCL-3, CCL-4, CCL-7, CXCL6, MIP-2, sCD27, sCD40, sCD40L, according to the manufacturer's instruction. Data in the scatter plots represent the mean of two technical replicates.

**In vitro stimulation and intracellular cytokine staining**. For functional assays on cytokine production by T cells, thawed isolated PBMCs were stimulated for 16 h at 37 °C in a 5% CO$_2$ atmosphere with anti-CD3/CD28 (1 μg/mL) in complete culture medium (RPMI 1640 supplemented with 10% fetal bovine serum and 1% each of l-glutamine, sodium pyruvate, nonessential amino acids, antibiotics, 0.1 M HEPES, 55 μM β-mercaptoethanol). For each sample, at least 2 million cells were left unstimulated as negative control, and 2 million cells were stimulated. All samples were incubated with a protein transport inhibitor containing brefeldin A (Golgi

Plug, Becton Dickinson) and previously titrated concentration of CD107a-PE. After stimulation, cells were stained with LIVE-DEAD Aqua (ThermoFisher Scientific) and surface mAbs recognizing CD3 PE- Cy5, CD4 AF700, and CD8 APC-Cy7 (Biolegend, San Diego, CA, USA). Cells were washed with stain buffer, and fixed and permeabilized with the cytofix/cytoperm buffer set (Becton Dickinson) for cytokine detection. Cells were next stained with previously titrated mAbs recognizing IL-17 BV421, TNF BV605, IFN-γ FITC, IL-2 APC, or granzyme-B BV421 (all mAbs from Biolegend). Then, a minimum of 100,000 cells per sample were acquired on a Attune NxT acoustic cytometer (ThermoFisher).

**Proliferation assay**. Cells from five patients and five controls were stimulated for 6 days in resting conditions, or after stimulation with anti-CD3 plus anti-CD28 mAbs (1 μg/mL each, Miltenyi Biotech, Bergisch Gladbach, Germany) and with 20 ng/mL IL-2. The fluorescent dye 5,6-carboxyfluorescein diacetate succinimidyl ester was used at a concentration of 1 μg/mL (TechnoFisher) according to standard procedures. Flow cytometric analyses for the identification of cycling cells belonging to different T cell populations were performed by gating TN, TCM, TEM, TE among CD4+ and CD8+ T cells[16].

**Mitochondrial bioenergetics and metabolic assays**. Due to lack of cells, we could only purify and study CD4+ T cells in five patients, compared with five controls. CD4+ T cells were first sorted using magnetic antibodies (Miltenyi Biotech), according to the manufacturer's instructions, seeded in triplicate at concentration of minimum $3.5 \times 10^5$ cells/well, rested for 4 h and stimulated for 16 h with anti-CD3 plus anti-CD28 as indicated above. Real-time measurement of OCR and ECAR was performed on a XFe-96 Seahorse Extracellular Flux Analyzer (Agilent, Santa Clara, CA, USA) using the MitoStress kit, according to manufacturer's procedures. OCR was measured in XF media (non-buffered DMEM medium, containing 10 mM glucose, 2 mM L-glutamine, and 1 mM sodium pyruvate) under basal conditions and in response to 2 μM oligomycin, 0.5 μM of carbonylcyanide- 4-(trifluoromethoxy)-phenylhydrazone (FCCP) and 0.5 μM of antimycin and rotenone (all from Sigma Aldrich). Indices of mitochondrial respiratory function were calculated from OCR profile: basal OCR (before addition of oligomycin), ATP-linked OCR (calculated as the difference between basal OCR rate and oligomycin-induced OCR rate), and maximal OCR (calculated as the difference of FCCP rate and antimycin plus rotenone rate)[53]. Basal and maximal ECAR values were also measured. OCR and ECAR values were normalized to the number of cells per well.

**Statistical analysis**. High-dimensional cytometric analysis was performed by using differential discovery in high-dimensional cytometry via high-resolution clustering[54]. Quantitative variables were compared using Mann–Whitney test. Simplified Presentation of Incredibly Complex Evaluation (SPICE) software (version 6, kindly provided by Dr. Mario Roederer, Vaccine Research Center, NIAID, NIH, Bethesda, MD, USA) was used to analyze flow cytometry data on T cell polyfunctionality[17]. Data are represented as individual values, means, and standard errors of the mean. Statistical analyses were performed using Prism 6.0 (GraphPad Software Inc., La Jolla, USA).

**Reporting summary**. Further information on research design is available in the Nature Research Reporting Summary linked to this article.

## Data availability

The source data underlying Figs. 1–9 are provided as a Source Data file. Original.fcs files concerning cytofluorimetric analysis (Figs. 1–4) are deposited at the flowrepository.org[55] in the following folders: T cell characterization: https://flowrepository.org/id/FR-FCM-Z2N5; T cell phenotype: https://flowrepository.org/id/FR-FCM-Z2N4. Source data are provided with this paper.

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

## Acknowledgements

S.D.B. and L.G. are Marylou Ingram Scholar of the International Society for Advancement of Cytometry (ISAC) for the period 2015–2020 and 2020–2025, respectively. We gratefully acknowledge Drs. Paola Paglia (ThermoFisher Scientific, Monza, Italy), Leonardo Beretta (Beckman Coulter, Milan, Italy), Emma Di Capua and Alfredo Caro-Maldonado (Agilent Technologies, Italy), for their support in providing reagents and materials in this study, and for precious technical suggestions. We also acknowledge Professor David W. Galbraith (Univ. of Tucson, AZ) for his help in revising the paper. This study was partially supported by unrestricted donations from Glem Gas spa (San Cesario, MO, Italy), Sanfelice 1893 Banca Popolare (San Felice S.P., MO, Italy) and Rotary Club Distretto 2072 (Clubs: Modena, Modena L.A. Muratori, Carpi, Sassuolo, Castelvetro di Modena), C.O.F.I.M. spa & Gianni Gibellini, Franco Appari, Andrea Lucchi, Federica Vagnarelli, Biogas Europa Service & Massimo Faccia, Pierangelo Bertoli Fans Club and Alberto Bertoli, Maria Santoro, Valentina Spezzani and BPER Banca. Finally, a special thank to the patients who donated blood to participate to this study.

## Author contributions

S.D.B., L.G.i., C.B., R.B., L.F., A.I., D.L.T., M.M.a. and A.P.a. carried out experiments and drafted the figures; L.G.o. drafted the tables.; M.M.e.s., M.M.e.n., J.M., G.F., R.F., R.T., M.S.i., L.B., F.F., A.P.i., E.C., M.G., G.G. and C.M. followed patients; L.C., M.S.a. and T.T. contributed to experimental procedures; S.D.B., L.G.i. and A.C. performed bioinformatic and statistical analyses; S.D.B., L.G.i., C.M. and A.C. conceived the study. All authors read and approved the paper.

## Competing interests

The authors declare no competing interests.
