## [Peer Review File · Nature Communications]

REVIEWER COMMENTS

Reviewer #1 (Viral responses, type 17 responses)(Remarks to the Author):

1. The authors Sara De Biasi et al. describe the characteristics of T cell compartment, plasma cytokines and cells producing cytokines in patients affected by Covid-19. They found that patients show increased amounts of CD4+ T cells that were activated, exhausted, stem memory or Treg and CD8+ T cells that were activated and exhausted. Covid-19 showed a dramatic increase of TH1, TH2, TH17 and Treg cytokines, chemokines, and galectins. Intracellular staining for cytokines after stimulus with anti-CD3/CD28 mAbs revealed a high capacity to produce a variety of molecules, including TNF- α , IFN- γ and IL-2 and IL-17. These findings are fairly informative for the immune responses in Covid-19 patients. My comments are as follows:

Comments

1. Studies have been conducted on a relatively small amount of subjects: maybe too small to draw generalized conclusions.

2. The studies show contradictory results for immunoactivation and immunosuppression in Covid-19 patients.

The studies confirm the importance of a massive immune activation in Covid-19 patients showing the increased plasma levels of inflammatory cytokines such as various chemokines, IFN- γ , Galectin-1, -3, -9, IL-1 α , IL-6, , IL-8, IL-15, IL-17, and TNF- α . Furthermore, Covid-19 patients expressed higher amounts of effector memory cells expressing CD45RA (EMRA), activated cells (co-expressing HLA-DR and CD38) within CD4+ T cells and significantly more terminally differentiated cells expressing CD45RA, much more activated cells (expressing HLA-DR and CD38) within CD8+T cells. CD4+ T cells from Covid-19 patients were able to produce significantly higher amounts of TNF- α , CD107a, IFN- γ , IL-2 and especially IL-17A and CD8+ T cells also produced significantly higher production of CD107a, IL-17A and especially IL-2 in response to stimulation with anti-CD3/CD28 in vitro. Thus, it is apparent that Covid-19 patients show massive immune activation, which is important not only for resolution of viral infection but also for autoimmune tissue damage resulting in acute respiratory distress syndrome (ARDS)

On the other hand, patients with Covid-19 also show anti-inflammatory responses: IL-10, and PD-L1, were markedly higher in plasma of Covid-19 patients who had more senescent/exhausted cells (PD1+,CD57+) and even more regulatory T cells (Treg) in CD4+T cells.. significantly more exhausted/senescent cells (PD1+, CD57+) in CD8+ T cells. Thus, the patients display an exhausted T cell compartment and increased amounts of Treg cells, which may prevent lung and other organs from autoimmune tissue damage.

In the discussion section, authors should discriminate massive immune activation (cytokine storm and T cell activation) from immunosuppression (anti-inflammation responses such as IL-10 production and increased amounts of Treg cells and exhausted T cells). Balance of immunoactivation and immunosuppression is critical for course of COVID-19 pneumonia.

For example,

“Elevated levels of serum proinflammatory cytokines and chemokines are known to contribute as “cytokine storm” to increased severity of disease caused by some strains of corona virus. Unique immunoregulatory system mediated by T cell exhaustion and suppressive cytokines such as IL-10 are responsible for limiting excessive inflammation and play an important role in homeostasis in the lungs. A balance in the levels of immunoactivation and immunosuppression may be crucial in host defense against highly pathogenic corona virus infection.”

“Exhausted T cells show phenotypic features of an exhausted state including the upregulated

expression of the inhibitory receptors programmed death (PD)-1. T cell exhaustion is characterized by functional unresponsiveness and prevent massive immunoactivation in order to prevent autoimmune tissue damage.”

3. The authors strongly state how this study impacts this research area as Th17 responses. Therefore, in addition to IL-8, authors should discuss the role of IL-17 in ARDS caused by COVID-19.

For example, “IL-17A, a T cell-derived proinflammatory cytokine, was shown to be involved in the mobilization and bactericidal activity of neutrophils. Neutrophils play critical roles in host defense against various pathogens, especially extracellular bacteria but in acute lung injury caused by highly pathogenic viral infection.”

4. Data on SEB stimulation may be deleted from Figure 5. SEB as a superantigen activates T cells expressing specific Vbeta repertoire such as Vbeta5. It may not reflect polyclonal activation.

5. Line 270 grammatical error : Most were lymphopenic, and most did not required non invasive ventilation, indicating that at the time of blood collection the disease was not too advanced.

Reviewer #2 (T cell exhaustion, viral responses)(Remarks to the Author):

The authors studied multiple immune parameters of 21 patients with pneumonia caused by the SARS-CoV-2, and 13 age- and sex-matched healthy control individuals. Multiparameter flow cytometry and quantifications of serum proteins was performed. The authors show that lymphocytes upregulated activation markers and skewed towards certain CD4 cell subtypes, in comparison to healthy controls. Moreover, several serum proteins were abnormally elevated.

The authors make interesting novel observations. They deserve the merit of having been able to rapidly study a large number of immune parameters and make them available for the scientific community. However, there are substantial weaknesses in the design of the study, and there are many statements representing overinterpretations and premature conclusions. One of it is that “SARS-CoV-2 provokes, in a very fast time, a dramatically confused immune response”. It remains unclear whether this is really the case and what is actually meant by this.

The statement that the SARS-CoV-2 has “an unusually high pathogenicity” is questionable.

Although suggested by some promising observations, it is not known whether tocilizumab treatment is significantly improving the outcome of COVID-19 patients, and if yes in which type of patients and severity of the disease, with or without complications.

The authors write that they have “studied the importance of the functional differentiation of T cells towards TH1, TH2, or TH17.” However, they did not much study the importance of these cells. Rather, they characterized them, leaving the questions about their importance largely open. At this stage this limitation is obvious and there is no reason to blame anyone. Researchers must always first characterize before they can go deeper to try elucidated causes and distinguish them from consequences. The point is that the language must be chosen accordingly. It is important to avoid overinterpretation.

The manuscript is full of overinterpretation, particularly the discussion, also with regard to the interpretation of the literature. Furthermore, the authors make to tight links between markers and cell

functions, on numerous occasions. For example, PD-1 expression does not necessarily mean that the cells are exhausted, actually rather not in the acute phase of a disease like studied here where PD-1 expression primary reflects cellular activation. Combining PD-1 and CD57 is also questionable for assessing T cell exhaustion. Based on the data, it is not appropriate to conclude on "exhaustion" and "senescence". The proper wording is "PD-1 expression", and "CD57 expression", not more.

A major limitation is that the authors show percentages but do not calculate absolute numbers of the cell populations analyzed. At some instances they nevertheless make use of terms like "amount(s)" suggesting that they determined the numbers of cells. The lack of calculating cell numbers of the lineages and subsets studied is particularly problematic because the total cell (lymphocyte) counts are not always in the normal range in COVID-19 patients. For each result it should be known whether the numbers of the cell subset are indeed abnormally low (or high).

The characterization of CD4 T cell subtypes is incomplete, as some polarization subtypes were studied whereas others not. The authors' conclusion that anti-IL-17 will be beneficial is premature. As most other data, the authors simply show high levels of IL-17 in the studied COVID-19 patients. To which extent this may be harmful or not remains to be determined. Even though it is often argued that the immune response may be too strong in COVID-19 patients, this remains to be clarified. The vast majority of patients successfully clear SARS-CoV-2, indicating that the immune response is very often productive. To argue that some immune activities are harmful requires at least some meaningful comparisons. The comparison with healthy individuals is insufficient; it is not a surprise that patients have multiple signs of immune activation when compared to healthy persons. It remains unclear to which degree the described characteristics are typical for COVID-19, and / or typical for different disease susceptibility and evolution. A comparison of patients with different degrees of disease severity is one possibility. And in different phases of the disease, when things look good, often early, vs. during severe disease which usually becomes evident only after the first week of illness. I acknowledge that many of these aims cannot be accomplished in very short time, but one should nevertheless make attempts in this direction, and discuss the limitations of what can be concluded at this point. Did the authors find different results in patients that had light versus severe disease, and / or recovered easily versus not?

What is the evidence that soluble PD-L1 can inhibit immune responses?

Maybe I missed it but I did not see significant changes in TSCM cells.

Minor points:

In the results section, it is often not clear to which Figure (part) the description is referring to. Some Figure parts are not mentioned anywhere in the text.

The authors suggest that their findings are representative for "the earliest stages of the infection", which is not correct.

Some points remain unclear because of insufficient/incomplete explanation. For example, the authors write "we had over 14,000 deaths." Does it mean "Italy had over 14,000 deaths"? It is also recommended to write the date, instead of only writing "to date".

The word "relevant" is not justified in "...caused a relevant production of TNF-a, IFN-g and IL-2".

This study should also be considered: Thevarajan et al. Breadth of concomitant immune responses prior to patient recovery: a case report of non-severe COVID-19. Nat Med. 2020. 1–10. doi:10.1038/s41591-020-0819-2.

The manuscript should be revised for improving the use of the English language, to correct grammar and other mistakes. For example, “heat ma in panel 2D” should probably say “heat map in Figure 2D”.

Abbreviations may be explained.

Point to point response to referees (**that are in bold**).

First, we wish to thank the referees. We believe that the new experiments that we performed to acknowledge their comments, and the elimination of the overstatements have significantly improved the manuscript.

Reviewer #1 (Viral responses, type 17 responses)(Remarks to the Author):

1. The authors Sara De Biasi et al. describe the characteristics of T cell compartment, plasma cytokines and cells producing cytokines in patients affected by Covid-19. They found that patients show increased amounts of CD4+ T cells that were activated, exhausted, stem memory or Treg and CD8+ T cells that were activated and exhausted. Covid-19 showed a dramatic increase of TH1, TH2, TH17 and Treg cytokines, chemokines, and galectins. Intracellular staining for cytokines after stimulus with anti-CD3/CD28 mAbs revealed a high capacity to produce a variety of molecules, including TNF- α , IFN- γ and IL-2 and IL-17. These findings are fairly informative for the immune responses in Covid-19 patients.

My comments are as follows:

1. Studies have been conducted on a relatively small amount of subjects: maybe too small to draw generalized conclusions.

We have increased the number of patients in which cytokine storm was studied, added other patients for the analysis of chemokine receptors, master regulator genes, cell proliferation and mitochondria bioenergetics, and have deleted general conclusions.

2. The studies show contradictory results for immunoactivation and immunosuppression in Covid-19 patients. The studies confirm the importance of a massive immune activation in Covid-19 patients showing the increased plasma levels of inflammatory cytokines such as various chemokines, IFN- γ , Galectin-1, -3, -9, IL-1 α , IL-6, IL-8, IL-15, IL-17, and TNF- α . Furthermore, Covid-19 patients expressed higher amounts of effector memory cells expressing CD45RA (EMRA), activated cells (co-expressing HLA-DR and CD38) within CD4+ T cells and significantly more terminally differentiated cells expressing CD45RA, much more activated cells (expressing HLA-DR and CD38) within CD8+T cells. CD4+ T cells from Covid-19 patients were able to produce significantly higher amounts of TNF- α , CD107a, IFN- γ , IL-2 and especially IL-17A and CD8+ T cells also produced significantly higher production of CD107a, IL-17A and especially IL-2 in response to stimulation with anti-CD3/CD28 in vitro. Thus, it is apparent that Covid-19 patients show massive immune activation, which is important not only for resolution of viral infection but also fort autoimmune tissue damage resulting in acute respiratory distress syndrome ARDS. On the other hand, patients with Covid-19 also show anti-inflammatory responses: IL-10, and PD-L1, were markedly higher in plasma of Covid-19 patients who had more senescent/exhausted cells (PD1+,CD57+) and even more regulatory T cells (Treg) in CD4+T cells.... significantly more exhausted/senescent cells (PD1+, CD57+) in CD8+ T cells. Thus, the patients display an exhausted T cell compartment and increased amounts of Treg cells, which may prevent lung and other organs from autoimmune tissue damage.

We fully agree that the results seem contradictory, but this is what we have found. The aspects of cell activation and inhibition resemble those present during different phases of sepsis, hyperactivation and immunoparalysis, and we have discussed this.

In the discussion section, authors should discriminate massive immune activation (cytokine storm and T cell activation) from immunosuppression (anti-inflammation responses such as IL-10 production and increased amounts of Treg cells and exhausted T cells). Balance of immunoactivation and immunosuppression is critical for course of COVID-19 pneumonia.

For example,

a) "Elevated levels of serum proinflammatory cytokines and chemokines are known to contribute as "cytokine storm" to increased severity of disease caused by some strains of corona virus. Unique immunoregulatory system mediated by T cell exhaustion and suppressive cytokines such as IL-10 are responsible for limiting excessive inflammation and play an important role in homeostasis in the lungs. A balance in the levels of immunoactivation and immunosuppression may be crucial in host defense against highly pathogenic corona virus infection."

b) "Exhausted T cells show phenotypic features of an exhausted state including the upregulated expression of the inhibitory receptors programmed death (PD)-1. T cell exhaustion is characterized by functional unresponsiveness and prevent massive immunoactivation in order to prevent autoimmune tissue damage."

We thank the referee for this comment, and we have added and expanded these sentences in the discussion:

a) page 13: T cells show phenotypic features of an exhausted state including the upregulated expression of the inhibitory receptors such as PD1. T cell exhaustion is characterized by functional unresponsiveness, which prevents massive immunoactivation and associated autoimmune tissue damage. Thus, it could be possible that in Covid-19 patients the activation of these cells is followed not only by the lack of clonal expansion (as revealed by the decreased proliferation) but also by the production of molecules that cause inflammation.

b) page 15: Elevated levels of serum proinflammatory cytokines and chemokines are known to contribute, as a cytokine storm, to the increased severity of disease caused by some strains of corona virus. Unique immunoregulatory system mediated by T cell exhaustion and suppressive cytokines such as IL-10 are responsible for limiting excessive inflammation and play an important role in homeostasis in the lungs. A balance in the levels of immunoactivation and immunosuppression may therefore be crucial in host defense against highly pathogenic corona virus infection.

3. The authors strongly state how this study impacts this research area as Th17 responses.

Therefore, in addition to IL-8, authors should discuss the role of IL-17 in ARDS caused by COVID-19. For example, "IL-17A, a T cell-derived proinflammatory cytokine, was shown to be involved in the mobilization and bactericidal activity of neutrophils. Neutrophils play critical roles in host defense against various pathogens, especially extracellular bacteria but in acute lung injury caused by highly pathogenic viral infection."

We thank the referee for this comment, and we have added and expanded the role of IL-17 in Covid-19 patients. We wrote. "In peripheral blood, patients also showed low percentages of both CD4+ and CD8+ T cells expressing CCR6 and high levels of CD161, which is typical of TH17 and of mucosal associated invariant T (MAIT) cells, respectively. The loss of circulating

CD4+,CD161+,CCR6+ cells contributes to disease progression in macaques infected with the simian immunodeficiency virus, and it has been shown that these cells accumulate in the rectal mucosa, enhancing inflammation (21). In the same infection model, the percentage of IL-17 producing CD8+,CD161+ cells present in the lung can be 4-fold higher than that in peripheral blood (22). Furthermore, cells present in the lung were able to produce more IL-17 than those from peripheral blood. Taken together, these findings underline the importance of IL-17 in Covid-19, and likely could pave the way to novel therapeutic approaches based upon IL-17 blockage by biological drugs that are already available".

4. Data on SEB stimulation may be deleted from Figure 5. SEB as a superantigen activates T cells expressing specific Vbeta repertoire such as Vbeta5. It may not reflect polyclonal activation.

As suggested, data on SEB have been deleted and are no longer present in the paper.

5. Line 270 grammatical error : Most were lymphopenic, and most did not required non invasive ventilation, indicating that at the time of blood collection the disease was not too advanced.

We have corrected this and other grammatical errors.

Reviewer #2 (T cell exhaustion, viral responses)(Remarks to the Author):

The authors studied multiple immune parameters of 21 patients with pneumonia caused by the SARS-CoV-2, and 13 age- and sex-matched healthy control individuals. Multiparameter flow cytometry and quantifications of serum proteins was performed. The authors show that lymphocytes upregulated activation markers and skewed towards certain CD4 cell subtypes, in comparison to healthy controls. Moreover, several serum proteins were abnormally elevated. The authors make interesting novel observations. They deserve the merit of having been able to rapidly study a large number of immune parameters and make them available for the scientific community. However, there are substantial weaknesses in the design of the study, and there are many statements representing overinterpretations and premature conclusions. One of it is that "SARS-CoV-2 provokes, in a very fast time, a dramatically confused immune response". It remains unclear whether this is really the case and what is actually meant by this.

We thank the referee for the kind words related to our effort. As indicated, we have deleted overstatements like the one quoted above, and smoothed the discussion.

The statement that the SARS-CoV-2 has "an unusually high pathogenicity" is questionable.
This statement has been deleted.

Although suggested by some promising observations, it is not known whether tocilizumab treatment is significantly improving the outcome of COVID-19 patients, and if yes in which type of patients and severity of the disease, with or without complications.

We have briefly described this aspect in the introduction, and we quote a reference related to a webinar that Dr. Cossarizza gave for Science on April 30th, in which for the first time we showed the data concerning the reduction of mortality by tocilizumab. The webinar is available online. In the figure here below, taken from the lecture, the main results are indicated. The paper is actually

under the second revision "for minor modifications" (in the Lancet Rheumatology), and, if accepted, can be quoted instead of the webinar.

[Redacted]

The authors write that they have “studied the importance of the functional differentiation of T cells towards TH1, TH2, or TH17.” However, they did not much study the importance of these cells. Rather, they characterized them, leaving the questions about their importance largely open.

To answer this comment, we rapidly developed an assay to quantify chemokine receptors and lineage specifying transcription factors that are related to TH1, TH2, TH17 cell differentiation, and have set up a novel panel of 10 mAbs detecting the molecules quoted here below. For technical reasons, also due to the difficulty to purchase reagents in this period, we could not analyze ROR- γ t. The results are reported in new figures, i.e., 1G, 1H, 2G, 2H. In CD4+ T cells, patients displayed a lower percentage of cells expressing CCR6 or CXCR3, and of those co-expressing CCR6 and CD161, but higher percentages of CXCR4+ or CCR4+ cells; no differences were noted in the expression of T-bet or GATA3. Concerning the expression of chemokine receptors and of transcription factors among CD8+ T cells, we found that patients expressed lower percentages of CCR6+, CXCR3+, T-bet+ or CD161high cells, and of CXCR3+, T-bet+ or CCR6+, CD161+ lymphocyte; they also had higher percentages of cells expressing CCR4+, CXCR4 or GATA3. These findings are discussed.

At this stage this limitation is obvious and there is no reason to blame anyone. Researchers must always first characterize before they can go deeper to try elucidated causes and distinguish them from consequences. The point is that the language must be chosen accordingly. It is important to avoid overinterpretation.

We thank the referee for this comment, and for understanding the current situation. We think that our new data, even if performed on a limited number of patients, could help in the analysis of the general picture that emerges.

The manuscript is full of overinterpretation, particularly the discussion, also with regard to the interpretation of the literature.

We have tried to modify all overinterpretations.

Furthermore, the authors make to tight links between markers and cell functions, on numerous occasions. For example, PD-1 expression does not necessarily mean that the cells are exhausted,

actually rather not in the acute phase of a disease like studied here where PD-1 expression primary reflects cellular activation. Combining PD-1 and CD57 is also questionable for assessing T cell exhaustion. Based on the data, it is not appropriate to conclude on “exhaustion” and “senescence”. The proper wording is “PD-1 expression”, and “CD57 expression”, not more.

We have changed the expression in some part of the manuscript with "PD1 expression" and "CD57 expression", even if we think that CD57 remains a decent marker for cell senescence. However, the data that we have obtained on cell proliferation show that, at least in terminally differentiated CD4+ or CD8+ T cell, a lower proliferation index exists among T cells.

A major limitation is that the authors show percentages but do not calculate absolute numbers of the cell populations analyzed. At some instances they nevertheless make use of terms like “amount(s)” suggesting that they determined the numbers of cells. The lack of calculating cell numbers of the lineages and subsets studied is particularly problematic because the total cell (lymphocyte) counts are not always in the normal range in COVID-19 patients. For each result it should be known whether the numbers of the cell subset are indeed abnormally low (or high).

We have solved this major limitation, as we could find (and use) in the patients' files the values of the absolute count of lymphocytes, and thus we have re-analyzed the data. The data are now reported, individually for each patient, in the new Figures 1B, 1C, 2B, 2C and in Table 1. The word "amount(s)" has been changed with "percentage" or "absolute number" where needed, in all the text.

The characterization of CD4 T cell subtypes is incomplete, as some polarization subtypes were studied whereas others not.

We fully agree with the referee, and indeed we have tried to better characterize CD4+ T cells.

The authors' conclusion that anti-IL-17 will be beneficial is premature. As most other data, the authors simply show high levels of IL-17 in the studied COVID-19 patients. To which extent this may be harmful or not remains to be determined. Even though it is often argued that the immune response may be too strong in COVID-19 patients, this remains to be clarified. The vast majority of patients successfully clear SARS-CoV-2, indicating that the immune response is very often productive. To argue that some immune activities are harmful requires at least some meaningful comparisons. The comparison with healthy individuals is insufficient; it is not a surprise that patients have multiple signs of immune activation when compared to healthy persons. It remains unclear to which degree the described characteristics are typical for COVID-19, and / or typical for different disease susceptibility and evolution. A comparison of patients with different degrees of disease severity is one possibility. And in different phases of the disease, when things look good, often early, vs. during severe disease which usually becomes evident only after the first week of illness. I acknowledge that many of these aims cannot be accomplished in very short time, but one should nevertheless make attempts in this direction, and discuss the limitations of what can be concluded at this point.

We are collecting now samples from patients with different degree of severity to better clarify this aspect, but this will take some time. This aspect has been discussed in the text, along with the limitations of our study: "We are well aware of the limitations of our study. First of all, the fact

that the relatively low number of patients, who were however chosen on the basis of similar clinical characteristics, does not allow us to divide them into those with a mild or severe course of the infection, having sufficient statistical power for further analyses. Second, due to fact that the study started when there was no treatment at all, and that in the days following blood collection some patients would have been treated and others not, it is difficult to understand which immune modifications can be considered predictive markers of the natural containment of the infection, or of the success of the therapy. Third, for this study we could not provide longitudinal data, but just compared the cohort with healthy donors."

Did the authors find different results in patients that had light versus severe disease, and / or recovered easily versus not?

We have indeed longitudinal data from a few patients, and it seems that, in comparison with patients who died or have a very long hospitalization (over one month, indicated as "BAD") those who recovered in a short time ("GOOD") had a higher amount (either percentage or absolute number) of naive CD8+ T cells, but not of any other parameter (see figure here below). However, since in both groups some of these patients were treated and some were not, and different drugs were used, it is difficult to establish the meaning of this observation.

What is the evidence that soluble PD-L1 can inhibit immune responses?

As requested, we have changed this sentence, and wrote: "We have found significantly high plasma levels of PD-L1 in our patients, and studies are needed to understand whether and how the infection uses this pathway for inhibition of an efficient antiviral immune response".

Maybe I missed it but I did not see significant changes in TSCM cells.

We have corrected this mistake and indicated that there were no differences in TSCM among CD4+ or CD8+ T cells in patients vs. control.

Minor points:

In the results section, it is often not clear to which Figure (part) the description is referring to. Some Figure parts are not mentioned anywhere in the text.

We have now mentioned all figure parts in the text.

The authors suggest that their findings are representative for “the earliest stages of the infection”, which is not correct.

We have deleted these words.

Some points remain unclear because of insufficient/incomplete explanation. For example, the authors write “we had over 14,000 deaths.” Does it mean “Italy had over 14,000 deaths”? It is also recommended to write the date, instead of only writing “to date”.

We have indicated the date (May 20th): "as of May 20th, Italy has counted more than 32,000 deaths amongst over 225,000 infections".

The word “relevant” is not justified in “....caused a relevant production of TNF-a, IFN-g and IL-2”.

We have changed the word relevant: ".. increased production"

This study should also be considered: Thevarajan et al. Breadth of concomitant immune responses prior to patient recovery: a case report of non-severe COVID-19. Nat Med. 2020. 1–10. doi:10.1038/s41591-020-0819-2.

The study has been considered and quoted (Reference N. 20)

The manuscript should be revised for improving the use of the English language, to correct grammar and other mistakes. For example, “heat ma in panel 2D” should probably say “heat map in Figure 2D”.

The manuscript has been kindly corrected word for word by a colleague of native English speakers (Professor David W. Galbraith, Univ. of Tucson, Arizona), who has been acknowledged.

Abbreviations may be explained.

Abbreviations are now explained in the text.

REVIEWERS' COMMENTS:

Reviewer #1 (Remarks to the Author):

The authors revised their manuscript in line with the referees' comments.
I have no further comments.

Reviewer #2 (Remarks to the Author):

The revision is well done.

The authors refer to a retrospective study with tocilizumab, and report of better results in tocilizumab treated COVID-19 patients as compared to controls. However, there was an increased prevalence of severe infection in the tocilizumab group. I suggest that the authors mention this point, and also the fact that the study was not randomized, with the notion that randomized studies are necessary to determine the real potential of tocilizumab treatment.

Otherwise I have no further comments and thank the authors for their consideration.

Point-to-point reply to the referees' comments

Reviewer #1 (Remarks to the Author):

The authors revised their manuscript in line with the referees' comments. I have no further comments.

We thank the reviewer for the consideration.

Reviewer #2 (Remarks to the Author):

The revision is well done.

The authors refer to a retrospective study with tocilizumab, and report of better results in tocilizumab treated COVID-19 patients as compared to controls. However, there was an increased prevalence of severe infection in the tocilizumab group. I suggest that the authors mention this point, and also the fact that the study was not randomized, with the notion that randomized studies are necessary to determine the real potential of tocilizumab treatment.

Otherwise I have no further comments and thank the authors for their consideration.

As requested, we have modified the sentence (last one in the Introduction) indicating that the study was retrospective and not randomized, as follows:

A retrospective study performed by our group evidenced that a drug able to block the biological activities of IL-6, such as tocilizumab, can significantly reduce invasive mechanical ventilation or death in severe COVID-19 pneumonia (defined as the concomitant presence of a respiratory rate ≥ 30 breaths per minute, blood oxygen saturation $\leq 93\%$, a $\text{PaO}_2/\text{FiO}_2$ ratio < 300 mmHg in room air and lung infiltrates $> 50\%$ of the lung, filed within 24-48 hours) (13). Even if our results need to be confirmed by randomized trials, it is to note that, in comparison with the control group, COVID-19 patients treated with tocilizumab had an increased prevalence of severe infection. This not only underlines the efficacy of a drug that contrasts cytokine storm, but also suggests the clinical evaluation of novel strategies, based upon the inhibition of the IL-17 pathway.

Reference 13 is the paper in press: Guaraldi G. et al., Tocilizumab in patients with severe COVID-19: a retrospective cohort study. The Lancet Rheumatology, 2020 in press.